# New Particle Formation in the Tropical Free Troposphere during CAMP²Ex: Statistics and Impact of Emission Sources, Convective Activity, and Synoptic Condition

Qian Xiao[1], Jiaoshi Zhang[1], Yang Wang[2], Luke D. Ziemba[3], Ewan Crosbie[3,4], Edward L. Winstead[3], Claire E. Robinson[3], Joshua P. DiGangi[3], Glenn S. Diskin[3], Jeffrey S. Reid[5], K. Sebastian Schmidt[6], Armin Sorooshian[7,8], Miguel Ricardo A. Hilario[8], Sarah Woods[9], Paul Lawson[9], Snorre A. Stamnes[3], Jian Wang[1]

[1]Department of Energy, Environmental and Chemical Engineering, Washington University in St. Louis, St. Louis, MO 63130, USA
[2]Department of Chemical, Environmental and Materials Engineering, University of Miami, Coral Gables, FL 33124, USA
[3]NASA Langley Research Center, Hampton, VA 23666, USA
[4]Science Systems and Applications, Inc., Hampton, VA 23666, USA
[5]Marine Meteorology Division, Naval Research Laboratory, Monterey, CA, USA
[6]Laboratory for Atmospheric and Space Physics, University of Colorado, Boulder, CO 80309, USA
[7]Department of Chemical and Environmental Engineering, University of Arizona, Tucson, AZ, 85721, USA
[8]Department of Hydrology and Atmospheric Sciences, University of Arizona, Tucson, AZ 85721, USA
[9]Stratton Park Engineering Company (SPEC), Boulder, CO 80301, USA

*Correspondence to*: Jian Wang (jian@wustl.edu)

**Abstract.**

Nucleation in the free troposphere (FT) and subsequent growth of new particles represent a globally important source of cloud condensation nuclei (CCN). Whereas new particle formation (NPF) has been shown to occur frequently in the upper troposphere over tropical oceans, there have been few studies of NPF at lower altitudes. In addition, the impact of urban emissions and biomass burning on the NPF in tropical marine FT remains poorly understood. In this study, we examine NPF in the lower and mid troposphere (3-8.5 km) over tropical ocean and coastal region using airborne measurements during the recent Cloud, Aerosol and Monsoon Processes Philippines Experiment (CAMP²Ex). NPF was mostly observed above 5.5 km and coincided with elevated relative humidity (RH) and reduced condensation sink (CS), suggesting that NPF occurs in convective cloud outflow. The frequency of NPF increases with altitude, reaching ~50% above 8 km.  An abrupt decrease in NPF frequency coincides with early monsoon transition, and is attributed to increased CS resulting from reduced convective activity and more frequent transport of aged urban plumes. Surprisingly, a large fraction of NPF events in background air were observed in the early morning, and the NPF is likely made possible by very low CS despite low actinic flux. Convectively detrained biomass burning plume and fresh urban emissions enhance NPF as a result of elevated precursor concentrations and scavenging of pre-existing particles. In contrast, NPF is suppressed in aged urban plumes where the reactive precursors are mostly consumed while CS remain relatively high. This study shows strong impact of urban and biomass burning emissions on the NPF in tropical marine FT. The results also illustrate the competing influences of different variables and interactions among anthropogenic emissions, convective clouds, and meteorology, which lead to NPF under a variety of conditions in tropical marine environment.

## 1 Introduction

New particle formation (NPF), the process of gas to particle nucleation and early growth to 2-3 nm, has been observed in many regions and over a wide range of altitudes, i.e., from the pristine to heavily polluted environment, from the tropics to the Arctic, and from boundary layer (BL) to tropopause layer (TL) (Twohy et al., 2002; Dada et al., 2017; Andreae et al., 2018; Kerminen et al., 2018; Zheng et al., 2021; Reid et al., 2016; Artaxo et al., 2022). Modelling studies suggest that on a global average, NPF contributes approximately half of the cloud condensation

nuclei (CCN) in the troposphere (Gordon et al., 2017), therefore strongly influencing cloud formation and climate (Kulmala et al., 2014). The rate of NPF depends on the concentrations of low volatility vapors (e.g., $H_2SO_4$ and highly oxygenated organics) that participate in the NPF, and the rate is also a strong function of temperature. As these low volatility vapors are mostly formed by photochemistry, their concentrations depend on the intensity of solar radiation as well as the concentration of precursors. Essentially all long-term surface measurements show that

the average solar radiation intensity is stronger during NPF event days compared with non-event days. Pre-existing aerosol particles serve as both a condensational sink for the low volatility vapors and a coagulation sink for newly formed particles, therefore they are expected to inhibit NPF. Indeed, observations at many locations have shown that NPF events in the troposphere typically occur under clean conditions (Kerminen et al., 2018; Kuang et al., 2009).

Over the oceans, NPF is typically observed in the free troposphere (FT). It had been long thought that NPF rarely

occurs within the remote marine boundary layer (MBL), because primary sea spray aerosols (SSA) present large condensation and coagulation sinks (Pirjola et al., 2000). A recent study shows that NPF takes place regularly in the upper part of the decoupled MBL following the passage of cold fronts over mid-latitude ocean, due to the combination of low existing aerosol loading, cold temperature, availability of reactive gases, and high actinic fluxes in the clear regions between scattered cumulus clouds (Zheng et al., 2021). In the BL over coastal regions, NPF can

occur in continental outflow such as transported urban plumes (Reid et al., 2016). In the FT over tropical and mid-latitude oceans, NPF was mostly observed in the air mass processed by convective clouds (Clarke et al., 1998; Clarke et al., 1999; Perry and Hobbs, 1994; Williamson et al., 2019). Intense NPF in convective outflow regions was observed in the tropical upper troposphere (UT) over both Pacific and Atlantic oceans (Williamson et al., 2019). Chemical-transport model simulations indicate this NPF in the tropical UT is a globally important source of CCN in

the lower troposphere. In the outflow of convective clouds, existing particles are depleted due to wet scavenging, leading to low condensation and coagulation sinks that promote NPF. At the same time, reactive gases such as dimethyl sulfide (DMS) are transported from the MBL to the outflow region, where the actinic flux is high and the reactive gases react to form low volatility species that participate in NPF (Williamson et al., 2019). Concurrent observations of elevated $H_2SO_4$ vapor concentration with the newly formed particles over open oceans suggest that

$H_2SO_4$ formed from oxidation of DMS likely plays an important role in NPF. While $NH_3$ and highly oxygenated compounds (HOM) can participate in NPF, modeling studies have shown that on a global average, about 80% of NPF between 5.8 km altitude and the top of the troposphere involves only sulfuric acid and water (binary nucleation; Gordon et al., 2017), and a large fraction of the NPF is ion-induced, especially over oceans where the overall NPF rate is relatively low (Dunne et al., 2016; Gordon et al., 2017). In addition to cloud outflow regions, newly formed

particles were also observed in the FT near the edge of cumulus clouds with enhanced actinic flux (Wehner et al.,

2015), and in continental outflow just above the BL cloud top (i.e., lower FT) over the northwestern Atlantic (Corral et al., 2022) and northeastern Pacific (Dadashazar et al., 2018).

Previous studies have greatly advanced our understanding of NPF in the marine environment. Over tropical oceans, most previous studies focused on the NPF in the UT, whereas the observations of NPF in the middle FT (i.e., 4-8 km) remain scarce (Clarke et al., 1998; Williamson et al., 2019). Kirkby et al. (2011) found that ion-induced binary nucleation associated with galactic cosmic ray can occur in the mid FT but is negligible in the BL, while the strongest aerosol formation takes place in the UT over tropic oceans (Kazil et al., 2006). In addition, previous measurements were mostly carried out in pristine environments. As a result, the impact of anthropogenic emissions on NPF in the tropical marine FT remains unclear. In this study, we take advantage of comprehensive airborne measurements during the Cloud, Aerosol and Monsoon Processes Philippines Experiment (CAMP$^2$Ex) to investigate NPF from the lower (~3 km) to upper FT (~8.5 km) in both background air masses and those impacted by urban emissions and biomass burning. The monsoon transition took place during the CAMP$^2$Ex campaign, providing an excellent opportunity to examine the impact of both changing air mass origins and convective activity on NPF. Through both statistical analysis and case studies, we quantify the frequency of NPF and the conditions under which NPF occurs in different air masses and their dependence on altitude. These results help improve the understanding of NPF in tropical marine environments, both in background conditions and under the influence from anthropogenic emissions and biomass burning.

## 2 Methodology

### 2.1 CAMP$^2$Ex and measurements used in this study

The CAMP$^2$Ex campaign, with the objective of characterizing the role of anthropogenic and natural aerosols in aerosol-cloud interaction in the vicinity of Philippines, included deployments onboard both the NASA P-3B aircraft and SPEC Learjet 35A (Reid et al., 2023). All data analyzed in this study are from the NASA P-3B aircraft, which flew 19 research flights from 24 August to 5 October 2019, covering South China Sea (SCS), Sulu Sea, West Pacific, and the continental FT (Fig. 1). The CAMP$^2$Ex campaign provided an excellent dataset to investigate NPF from the lower to upper FT (3-8.5 km) in a range of air masses, including background air and those influenced by Borneo biomass burning smoke, Asian pollution, and local emissions from Philippines (Hilario et al., 2021).

The measurements examined in this study include aerosol properties, carbon monoxide (CO), methane (CH$_4$) and ozone (O$_3$) mixing ratios, meteorological parameters, and radiation (see Table 1 for details). Ambient aerosol was sampled by using a "Clarke" style forward facing shrouded solid diffuser that was operated iso-kinetically (Mcnaughton et al., 2007). Two condensation particle counters (CPCs, Model 3756 and 3772, TSI Inc.) measured the total number concentrations of particles nominally larger than ~3 and ~10 nm ($N_{>3\,nm}$ and $N_{>10\,nm}$), respectively. An additional CPC (TSI Model 3772) sampled downstream of a thermal denuder operated at 350 °C and provided non-volatile particle number concentration (nonvolatile $N_{>10\,nm}$). Aerosol size distributions were characterized by a fast integrated mobility spectrometer (FIMS, 10-600 nm; Wang et al., 2017a; Wang et al., 2017b; Wang et al., 2018)

and a laser aerosol spectrometer (LAS, Model 3340, TSI Inc., 100-3000 nm). The aerosol samples measured by FIMS and LAS were dried both actively by Nafion driers and passively due to higher aircraft cabin temperature than the ambient. Size distributions provided by LAS were size-corrected assuming a particle refractive index of ammonium sulfate (Moore et al., 2021).

The cloud droplet spectra were measured by a fast cloud droplet probe (FCDP, SPEC Inc.; Lawson et al., 2017).

Several trace gases measured in-situ onboard the P-3B were used to identify different air mass origins. CO and $CH_4$ mixing ratios were characterized by a dried-airstream near-infrared cavity ring-down absorption spectrometer (Model G2401-m, PICARRO Inc.; Digangi et al., 2021; Baier et al., 2020). $O_3$ was measured by a dual-beam UV adsorption sensor (Model 205; 2B Technologies; Digangi et al., 2021). Water vapor mixing ratio and relative humidity (RH) were provided by a diode laser hygrometer (DLH; Diskin et al., 2002; Podolske et al., 2003) at 1 Hz.

Upwelling and downwelling shortwave irradiance from 350-2150 nm were measured by the solar spectral flux radiometer (SSFR; Norgren et al., 2022; Schmidt et al., 2021; Chen et al., 2021).

**Table 1.** List of measurements used in this study, instruments, and sampling frequency.

| Parameter/Variable | Instruments/Methods | Sampling frequency |
| --- | --- | --- |
| Aerosol number concentration (> 3 nm) | Condensation particle counter (CPC, TSI-3756) | 1 Hz |
| Aerosol number concentration (> 10 nm) | Condensation particle counter (CPC, TSI-3772) | 1 Hz |
| Number concentration of non-volatile particles (> 10 nm) | Condensation particle counter (CPC, TSI-3772) downstream of a thermodenuder | 1 Hz |
| Aerosol size distribution (10-600 nm) | Fast integrated mobility spectrometer (FIMS) | 1 Hz |
| Aerosol size distribution (100-3000 nm) | Laser aerosol spectrometer (LAS, TSI-3340) | 1 Hz |
| Cloud droplet size distribution (2-50 µm) | Fast cloud droplet probe (FCDP, SPEC Inc.) | 1 Hz |
| Ozone mixing ratio | Dual cell broadband UV absorption spectrometry (2B Technologies, Model 205) | 0.4 Hz |
| CO and methane mixing ratio (dry mass fraction) | Near-IR cavity ringdown absorption spectroscopy (PICARRO Inc., G2401-M) | 0.4 Hz |
| Relative humidity with respect to water (RH) | Diode laser hygrometer (DLH, NASA Langley Research Center) | 1 Hz |
| Upwelling and downwelling shortwave irradiance | Solar spectral flux spectrometer (SSFR) | 1 Hz |

| Latitude/longitude/GPS altitude | Litton 251 | 1 Hz |
| Air temperature | Rosemont 102 Fast | 1 Hz |

When the P-3B was inside clouds, aerosol measurements were impacted by shattering of cloud droplets and/or ice
particles on the iso-kinetic aerosol sampling inlet. In-cloud periods and additional 3-second buffer time immediately
before and after the in-cloud periods were identified and flagged based on hydrometeor measurements (i.e., cloud
flag, available in CAMP²Ex data archive https://asdc.larc.nasa.gov/project/CAMP2Ex). Aerosol measurements
during the flagged periods were excluded from the analysis to minimize the influence of the measurement artifacts.
We calculated condensation sink (CS) from the ambient aerosol size distribution (Kulmala et al., 2012), which was
derived by combining dry particle size distribution measured by FIMS (10-600 nm) and LAS (600-1000 nm),
ambient RH, and an average hygroscopicity parameter ($\kappa$). Aerosol mass spectrometer (AMS) measurements show
that on average, $(NH_4)_2SO_4$ represents 90% of the $PM_1$ mass above 5 km, where vast majority of the NPF events
were identified. A $\kappa$ value of 0.53 was therefore applied to calculate particle hygroscopic growth factor at ambient
RH (Petters and Kreidenweis, 2007) for each size bin of the combined dry size distribution. All aerosol number
concentrations, size distributions, and CS are reported at standard temperature and pressure (1013.25 hPa and 273.15
K, STP). As there was no direct measurement of actinic flux, we calculated the ultraviolet (UV) irradiance by
integrating the irradiance over the wavelength range of 350-400 nm measured by the SSFR and used it as a proxy of
actinic flux. The total UV irradiances were derived by summing both upwelling and downwelling components.

**2.2 Identification of NPF events**

Ideally, the concentration of incipient particles (i.e., stable clusters with diameters between 1 to 2 nm) is used to
identify NPF events. However, given the challenges of measuring particle concentration below 2 nm onboard
research aircraft, many airborne studies have used the concentration ratio of the particles with diameter above 3 nm
($N_{>3\ nm}$) to that above 10 nm ($N_{>10\ nm}$) and/or the number concentration of particles between 3 nm and 10 nm ($N_{3-10\ nm}$) to characterize NPF events (Crumeyrolle et al., 2010; Zheng et al., 2021). In this study, we use $N_{>3\ nm}/N_{>10\ nm}$ to
identify NPF events, following a similar approach described by Zheng et al. (2021). A ratio ($N_{>3\ nm}/N_{>10\ nm}$)
substantially above 1 indicates the presence of newly formed particles between 3 and 10 nm, and thus a recent NPF
event. Considering the response times of the CPCs to step changes in particle concentration (i.e., about 2 seconds to
reach 90% of concentration step change), we first averaged the 1-second measurements of particle number
concentrations (i.e., $N_{>3\ nm}$ and $N_{>10\ nm}$) into 10-s intervals. For each of the 10-s intervals, the ratio of average $N_{>3\ nm}$
to average $N_{>10\ nm}$ and the uncertainty of the ratio ($\sigma_R$) were derived. New particles are considered to be present when
the ratio is above 1.3 plus three times of the uncertainty:

$$\frac{N_{>3\ nm}}{N_{>10\ nm}} > 1.3 + 3 \cdot \sigma_R \qquad (1)$$

An NPF event was identified when at least three consecutive 10-s intervals indicate the presence of newly formed particles. Given the P-3B flew at a speed of ~160 m/s, this translates into a minimum spatial scale of ~5 km. For the 105 NPF events identified, the durations range from 30 to 1150 s, corresponding to spatial scales of 5-196 km.

To contrast the conditions between NPF and non-NPF events, we also define non-NPF periods following a similar approach. Specifically, a non-NPF period consists of a minimum of 6 consecutive 10-s intervals (i.e., a minimum of 60 s in duration) with all the intervals showing the ratio of averaged particle concentrations (i.e., $N_{>3\,nm}/N_{>10\,nm}$) statistically below 1.05:

$$\frac{N_{>3\,nm}}{N_{>10\,nm}} < 1.05 - 3 \cdot \sigma_R \qquad (2)$$

We note the criteria for non-NPF periods are quite strict. Due to measurement counting statistics, some periods that are absent of newly formed particles might not be identified as non-NPF periods. Similarly, some weak NPF events might not be picked up by the criteria (i.e., Eq. (1)) described above either.

**2.3 K-means classification of NPF events**

To examine the conditions that lead to the observed NPF, we performed k-means clustering on the matrix consisting of event mean values of CS, RH, ambient temperature, and UV irradiance for 95 NPF events (10 events were excluded due to missing data for one or more of the four variables). RH was included as one of the variables because an elevated RH in the mid to upper FT often indicates air with more moisture and reactive gases (e.g., DMS) that are vertically transported by convective clouds from the lower atmosphere (Reid et al., 2019). In addition, elevated RH in the cloud outflow regions is typically associated with high concentration of water vapor, which has been shown to participate in binary nucleation (Vehkamäki et al., 2002). The total UV irradiance was included as a proxy for actinic flux due to the lack of direct measurement.

The four variables (i.e., RH, CS, ambient temperature, and UV irradiance) were first normalized using z-score standardization. The optimal number of clusters K was determined as six using the elbow method together with Silhouette coefficient (Syakur et al., 2018; Rousseeuw, 1987). We then performed the k-means clustering via MATLAB based on k-means ++ algorithm (Arthur and Vassilvitskii, 2007; Lloyd, 1982) and a prescribed setting of 5,000 iterations. Consequently, the 95 NPF events were classified into 6 clusters, and each cluster contains 7-35 NPF events. General statistics of the four key variables are presented for the six clusters in Table 2.

**Table 2. General statistics of key parameters for the 6 clusters identified using k-means classification.**

| Cluster # | Number of events | Amount of 1-s data | Mean±std altitude, m | Mean±std temperature, °C | Mean±std UV irradiance, W m$^{-2}$ | Mean±std RH, % | Mean±std CS, $10^{-3}$ s$^{-1}$ |
|---|---|---|---|---|---|---|---|
| 1 | 35 | 5550 | 6104.6±591.9 | -4.6±3.3 | 108.8±13.6 | 75.4±9.0 | 1.1±0.5 |
| 2 | 20 | 3870 | 7708.8±433.2 | -15.2±2.4 | 104.5±13.1 | 79.8±8.5 | 2.0±0.7 |
| 3 | 13 | 3960 | 6392.4±369.8 | -7.3±1.8 | 60.9±14.4 | 82.6±7.0 | 1.1±0.5 |
| 4 | 9 | 1190 | 7532.1±438.2 | -12.9±2.5 | 118.8±21.2 | 33.3±13.5 | 1.2±0.6 |
| 5 | 11 | 790 | 6698.5±650.7 | -10.3±4.2 | 93.7±23.6 | 61.3±6.3 | 5.1±1.2 |
| 6 | 7 | 400 | 3959.3±671.3 | 4.2±4.3 | 58.1±24.1 | 74.2±10.5 | 2.9±2.0 |

## 2.4 Classification of air mass types during CAMP²Ex

To investigate the impact of air masses on NPF, we classified air masses sampled during CAMP²Ex into four types: background air (background, hereafter), biomass burning (BB-influenced, hereafter), air mass influenced by urban (urban-influenced, hereafter) and mixed urban/biomass burning. The classification is largely based on the observed

ratios of enhancement in methane to that in CO ($\Delta CH_4/\Delta CO$), taking advantage of the relatively long lifetime of both trace gases in the FT (the details of the approach can be found at: https://doi.org/10.5067/Airborne/CAMP2Ex_TraceGas_AircraftInSitu_P3_Data_1). As reported by literature (Nara et al., 2017; Worden et al., 2017), low $\Delta CH_4/\Delta CO$ has been frequently observed in biomass burning plumes (typically < 10%), whereas much higher ratios (typically close to 100%) have been reported in fossil fuel

combustion emissions (Helfter et al., 2016). Here the term background is defined to differentiate the air masses from BB-influenced and urban-influenced air masses, and it does not strictly refer to very clean conditions. In this study, we investigate the impact of air masses on NPF by focusing on NPF events observed in background, biomass burning and urban-influenced air masses (i.e., the first three air mass types).

## 3 Overview of NPF events during CAMP²Ex

### 3.1 General statistics

There was a total of 19 research flights (RFs) during CAMP²Ex. Figure 1 shows the flight tracks and the locations where NPF in three major air mass types was observed. These RFs covered the ocean east and west of Luzon Island, and two of them (RF8 and RF18) sampled over Luzon Island and upwind/downwind of Metro Manila. The date and sampling areas of all RFs, together with the duration and key variables of the identified NPF events, are presented in

Table S1. These NPF events are about evenly distributed over the open ocean, coastal regions and the land. Most NPF events were observed above 5.5 km when RH exceeded 50%. Below 3 km, only a few short periods with elevated $N_{>3\,nm}/N_{>10\,nm}$ were observed within the BL about 50 kilometers downwind west of metro Manila during

RF18, and are closely associated with shipping and/or urban emissions. The elevated $N_{>3\,nm}/N_{>10\,nm}$ likely occurred immediately following the dilution of vehicle and engine emissions (e.g., Uhrner et al., 2011; Wehner et al., 2009), and they are not considered as NPF events and therefore excluded from further analyses. NPF frequency, defined as the ratio of the sampling time when new particles were observed to the total flight time, decreases drastically starting from RF11 on 19 September (Fig. S1). No NPF events were observed from RF12 through RF17, despite the flight tracks overlap with the earlier flights in terms of location and altitude range. This abrupt decrease in NPF frequency coincides with the early monsoon transition starting on 20 September (Hilario et al., 2021), indicating a strong impact of synoptic condition on NPF in this region that will be discussed in the next section (Section 3.2).

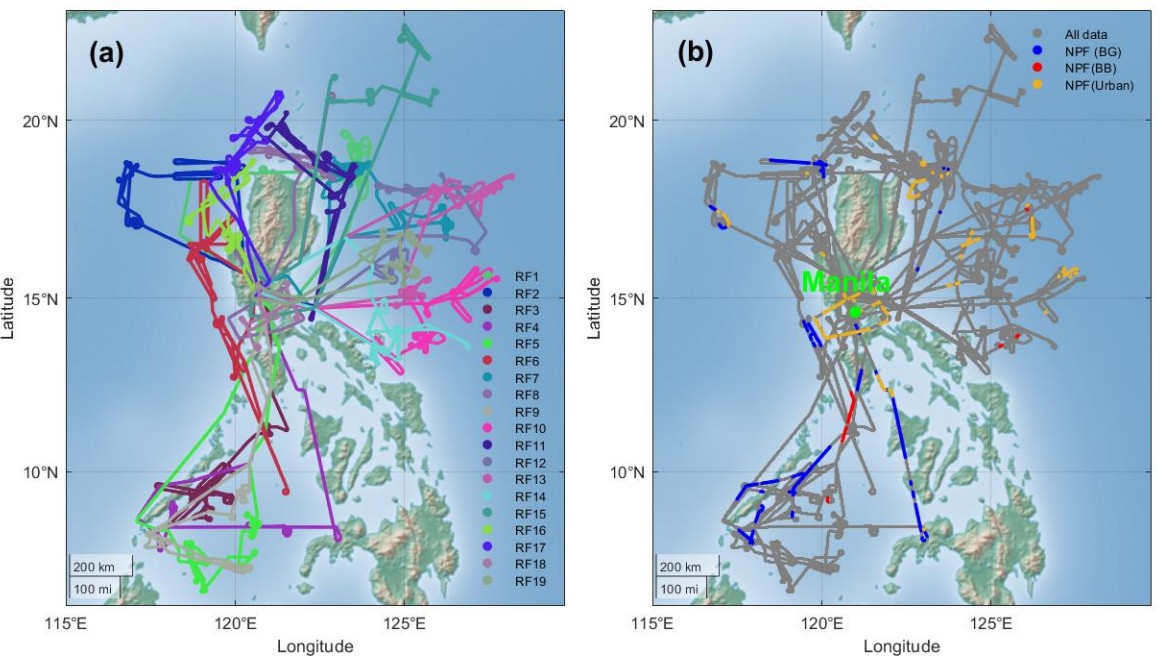

**Figure 1. (a) Flight tracks during CAMP$^2$Ex colored according to research flight number. (b) Locations of NPF events observed in background, biomass burning-influenced and urban-influenced air masses colored by the air mass type.**

**3.2 Vertical profile of NPF frequency in different air mass types**

For each of the three air mass types (i.e., background, urban-influenced, and BB-influenced), NPF frequency was calculated for 500 m altitude bins between 3 and 8.5 km and the vertical profiles of NPF frequency are examined. In addition, we compare the vertical profiles of the CS and RH between NPF and non-NPF periods (see Sect. 2.2 for the definitions of NPF and non-NPF periods). Note that due to the limited sampling, no non-NPF periods above 7.5 km are identified based on the criteria described in Sect. 2.2. For the comparison above 7.5 km, the non-NPF period

is instead defined as the entire period when P3-B sampled outside of clouds except when newly formed particles were observed (i.e., when Eq. (1) is satisfied).

Below 5.5 km, NPF frequency is very low (below 3%) and NPF was mostly observed in the urban-influenced air masses (Fig. 2a). NPF frequency shows strong increases with altitude above 5.5 km for all three air mass types,
reaching about 50% above 8 km. As CS is largely independent of altitude above 5.5 km (Fig. S2), the strong increase of NPF frequency is likely due to lower temperature and higher galactic cosmic rays ionization rate at higher altitudes, at least partially (Kazil et al., 2006). Figure 2b shows that over the entire altitude ranges examined, NPF consistently occurred with elevated RH, suggesting NPF in outflow regions and detrainment layers of convective clouds. This is confirmed by the flight video, and also consistent with earlier studies (Clarke et al., 1998;
Perry and Hobbs, 1994). Previous studies show that the mixing of air masses with different temperature and vapor concentrations can enhance nucleation rates (Khosrawi and Konopka, 2003; Nilsson and Kulmala, 1998; Nilsson et al., 2001; Wehner et al., 2010). Therefore, the mixing of cloud outflow and surrounding air may contribute to the observed NPF events. The relationship between NPF and CS shows an altitude dependence (Fig. 2c). Above 5.5 km, newly formed particles were observed with reduced CS with median values mostly below ~ 0.002 s$^{-1}$, comparable to
CS below $8 \times 10^{-4}$ s$^{-1}$ in the tropical mid-FT reported by Williamson et al., 2019). In contrast, NPF below 5.5 km coincides with elevated CS, where NPF events mostly took place in urban-influenced air masses. The altitude dependence of the relationships among air masses, CS, and NPF implies competing influences from different processes (i.e., production and removal of nucleating species), which will be further discussed in Sect. 4. We also compare the NPF frequency, CS and RH as a function of altitude between SWM and MT periods. The NPF
frequency during the MT is lower than that during SWM at most altitudes above 5.5 km (Fig. S3). The decrease of NPF frequency during the MT is likely due to increased CS (Fig. S3b), which may be a result of reduced convective activity as indicated by lower RH (Fig. S3c) and thus reduced wet scavenging. The more frequent long range transport of aged urban plumes may also contribute to the elevated CS during the MT (Hilario et al., 2021).

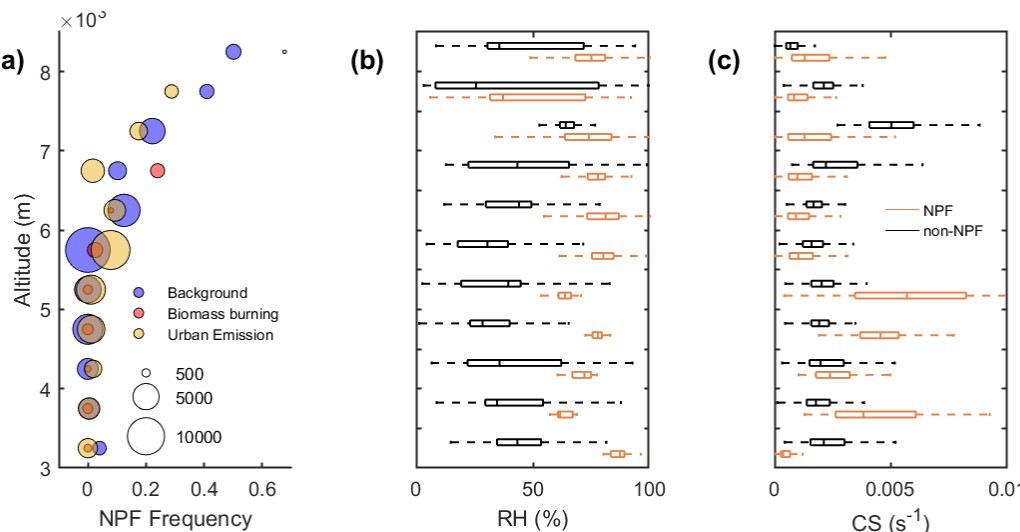

Figure 2. (a) The vertical profile of NPF frequency for three major air mass types. NPF Frequency is defined as the ratio of total duration of NPF period to the total sampling time outside of the clouds for each air mass

**type. Also shown are the comparison of (b) RH and (c) CS between NPF and non-NPF periods, where black denotes non-NPF and orange denotes NPF periods, respectively.**

### 3.3 K-means classification results

As described in Sect. 2, a total number of 95 NPF events were classified into six clusters based on CS, RH, temperature, and UV irradiance. Figure 3a shows the contribution of the three air mass types to each NPF event cluster and the mean sampling altitudes of the clusters. Clusters #1-3 represent the vast majority (i.e., 76%) of data collected during the NPF events. Cluster #1 consists mostly of NPF events associated with polluted air masses (i.e., BB-influenced or urban-influenced). In contrast, NPF events in clusters #2 and #3 were mostly observed in

background air. Altogether, clusters #4-6 represent 24% of the NPF event data, the majority of which was observed in urban-influenced air masses. Figure 3b-f show that new particles form under a wide range of conditions, and the formation exhibits varying intensities, as indicated by different $N_{>3\,nm}/N_{>10\,nm}$ and $N_{3\text{-}10\,nm}$ values (Fig. 3d-e). Most of the NPF events in clusters #1-4 were observed in air masses with CS lower than 0.002 s$^{-1}$, consistent with the findings from earlier studies (e.g., Williamson et al., 2019). The NPF events classified as cluster #5 have the highest

CS compared to the other clusters and were mostly observed during RF18 and 19. These two flights took place near the end of CAMP$^2$Ex during monsoon transition, when air mass origins and meteorological conditions are likely different from those of earlier flights. The potential mechanism for the NPF events with high CS will be discussed in Sect. 4.3. Figure 3b shows that most NPF occurred with high actinic flux (indicated indirectly by the UV irradiance), as in clusters #1, #3, and #4. However, cluster #2 NPF events occurred with much lower UV irradiance, which will

be discussed in Sect. 4.1. In terms of RH, except for cluster #4, all NPF clusters were observed with median RH above 50% and the RH is statistically higher than that during corresponding non-NPF periods (not shown), again indicating that NPF mostly takes place in air masses processed by convective clouds. NPF in cluster #4 occurred under the driest conditions (Fig. 3f) but with the highest UV irradiance (Fig. 3b), and $N_{3\text{-}10\,nm}$ is statistically the lowest among all clusters (Fig. 3e).

Because it takes some time for incipient particles to grow into 3-10 nm size range, the NPF events identified using $N_{>3\,nm}/N_{>10\,nm}$ value are likely several hours after the formation of the incipient particles. As the incipient particles are efficiently removed by coagulation inside clouds, we expect that air masses with elevated $N_{>3\,nm}/N_{>10\,nm}$ remained cloud free and did not experience precipitation since the recent particle formation. Therefore, CS and RH, which are among the NPF related variables examined in this study, are unlikely to vary drastically between the particle

formation and the observation of elevated $N_{>3\,nm}/N_{>10\,nm}$. NPF and subsequent particle growth can lead to an increase of CS. For elevated $N_{>3\,nm}/N_{>10\,nm}$ observed under conditions of low CS, the formation of incipient particles likely occurred with comparable or even lower CS. UV irradiance has a strong diurnal variation and depends on the cloud condition, and it can change substantially over a period of several hours. In this study, most NPF events (i.e., elevated $N_{>3\,nm}/N_{>10\,nm}$) were observed around noon time, and were under higher levels of UV irradiance compared

to the non-NPF periods at the same altitude, consistent with earlier studies (Kerminen et al., 2018) showing that solar radiation is generally higher during NPF event days than non-event days. This suggests UV irradiance at the time when elevated $N_{>3\,nm}/N_{>10\,nm}$ was observed is likely representative of that at the time of particle formation.

Some of the NPF events (i.e., cluster #2) were observed under conditions of low UV irradiance, and the potential mechanisms are discussed in Sect. 4.1.

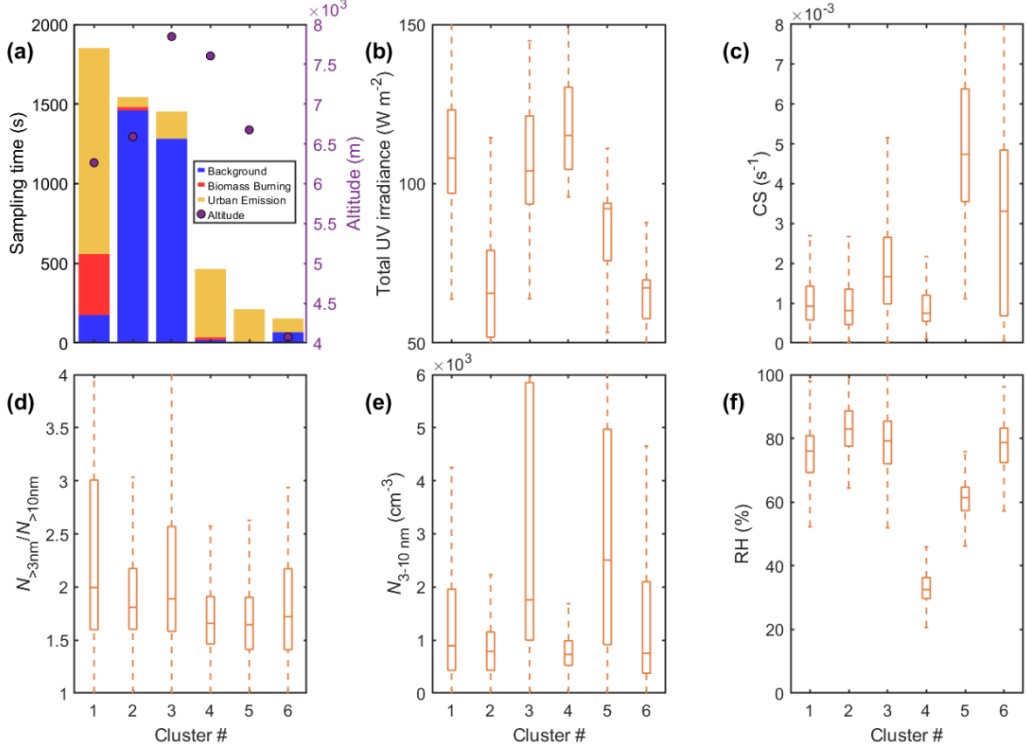


**Figure 3. (a) Number of 1-s data classified into each cluster and contributions of the three major air mass types. The other five panels compare the statistics of (b) total UV irradiance, (c) CS, (d) the ratio of number concentration of particle larger than 3 nm to that of particle larger than 10 nm ($N_{>3\ nm}/N_{>10\ nm}$), (e) number concentration of particles in the size range of 3-10 nm ($N_{3-10\ nm}$), and (f) RH of the 6 clusters.**

**4 Characteristics of NPF in different air mass types**

We combine the k-means classification (i.e., based on T, RH, CS and UV irradiance) and air mass classification to investigate the impact of both meteorological conditions and emissions on the NPF. We divide the above clusters into multiple types, including NPF in background, mid-altitude NPF in polluted air, high-altitude NPF in polluted air, etc. In the following sections, we will examine NPF of each type and investigate the conditions that lead to NPF

in different air masses as a function of altitude.

**4.1 NPF observed in the background air mass**

NPF in the background air (CO concentration < 110 ppbv and $CH_4$ concentration < 1.86 ppm) was mostly observed in the early part of the campaign (i.e., RF2-RF6) during the southwest monsoon phase. These NPF events, mostly classified as clusters #2 and #3 (Fig. 3a), took place from 6 km to 8.5 km over Sulu Sea/West Luzon.

We further divide background NPF events into two types based on the result of k-means classification: one classified into cluster #2 while the other classified into cluster #3 (mostly sampled during RF4 and RF6). The main differences between these two types include UV irradiance, temperature, and CS. Figure 4 compares key variables of the two background NPF types as a function of RH. For both types, RH is mostly in the range of 60-100% and concentrated between 70% and 90%. The high RH indicates that the background NPF took place in cloud processed air (e.g., outflow region or detrainment layers). UV irradiance, $N_{>3\ nm}/N_{>10\ nm}$, and $N_{3-10\ nm}$ show similar variations with RH (Fig.4b, e, and f), and exhibit the highest values in the RH range of 60-80% for both types. This suggests UV irradiance plays an important role in these background NPF events, in agreement with the findings of earlier studies (e.g., Perry and Hobbs, 1994). The high irradiance under RH between 60 and 80% is attributed to the presence of clouds, as confirmed from the videos recorded by the forward camera onboard the P-3B. The effect of UV irradiance on the background NPF is also consistent with an earlier study (Wehner et al., 2015) that reports newly formed particles in regions with enhanced UV irradiance near cumulus clouds. As RH increases above 80% and approaches 100%, the UV irradiance decreases from its peak values, accompanied by decreases in both $N_{>3\ nm}/N_{>10\ nm}$ and $N_{3-10\ nm}$. The decrease in UV irradiance above 80% RH is likely due to attenuation of solar radiation in the immediate vicinity of clouds and between cloud layers (Hamed et al., 2011). As it takes some time for the incipient particles to grow and reach detectable sizes (i.e., > 3 nm), the reduced $N_{>3\ nm}/N_{>10\ nm}$ and $N_{3-10\ nm}$ when RH is above 80% are probably due to a combination of reduced actinic flux and recently nucleated particles having not reached detectable sizes yet in the immediate vicinity of clouds.

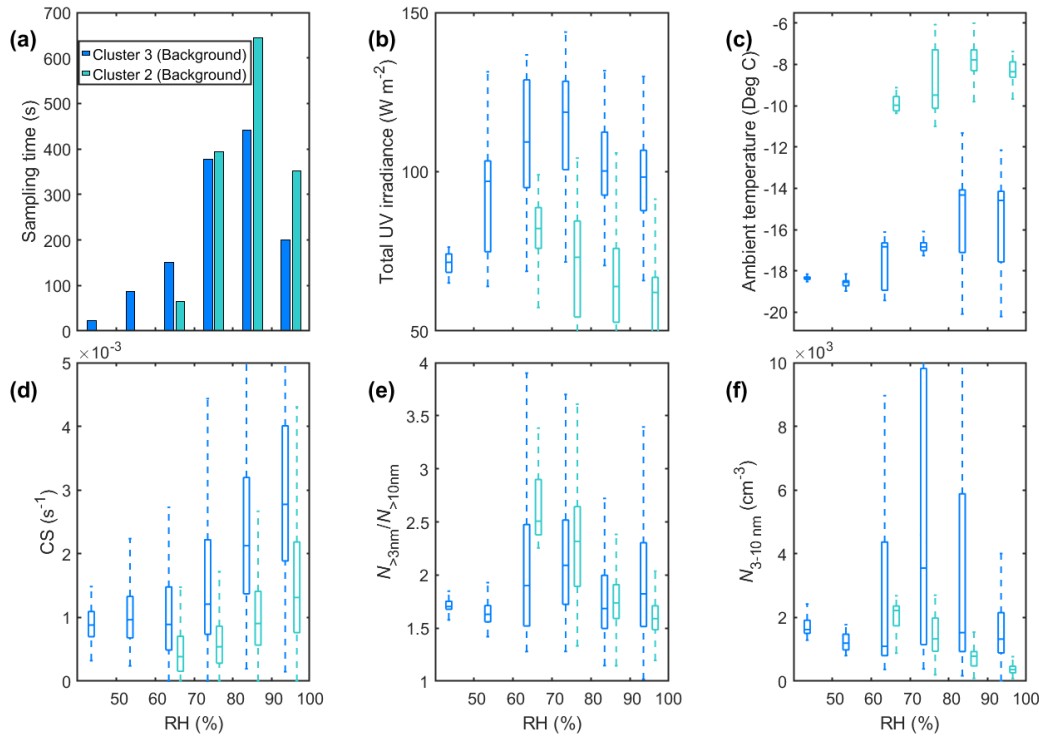


**Figure 4. (a) Sampling time, (b) UV irradiance, (c) ambient temperature, (d) CS, (e) $N_{>3\,nm}/N_{>10\,nm}$, and (f) $N_{3-10\,nm}$ as a function of RH for two types of background NPF classified into k-mean clusters #2 and #3, respectively. The statistics are shown for 10% RH bins from 40% to 100% (i.e., 40%-50%, 50%-60%, etc.)**

Whereas the two background NPF types share some similar features, there are also clear differences between them.

The background NPF events in cluster #3 occurred around noontime with high UV irradiance, in agreement with previous studies (Kerminen et al., 2018). In contrast, the background NPF in cluster #2 was mostly observed under conditions of low UV irradiance in the early morning, which has rarely been reported. We note that under clear sky, UV actinic flux has a weaker dependence on solar zenith angle (SZA). The UV actinic flux is estimated from the UV irradiance, SZA, and cloud condition (Supplementary Section 1). Both UV irradiance and actinic flux during the

morning background NPF events are statistically lower than those during the NPF events that occurred during 10:00-14:00 local time in the same altitude range (Fig. S4). The median UV irradiance during morning background NPF events is about 28% lower than that of the NPF events around noon, while the median UV actinic flux is about 11% lower. The morning background NPF occurred with some of the lowest CS (i.e., mostly below 0.001 s⁻¹) during CAMP2Ex, substantially lower than those during background NPF events around noontime (Fig. 4d).

One possible explanation for the background NPF in the early morning is that the new particles were formed the day before under high UV irradiance/actinic flux, survived scavenging overnight, and were detected in the morning. However, the very low CS condition (i.e., CS < 0.001 s⁻¹) is much more prevalent in the early morning than in the late afternoon (Fig. S5 and Supplementary Section 2). In addition, the frequency of NPF in the early morning is

about 20 times higher than that in the afternoon, suggesting that the observed new particles most likely formed in the

morning instead of the day before. The background NPF in the early morning is likely made possible by the much

lower CS despite the lower UV irradiance and actinic flux. We speculate the prevalence of very low CS condition in

the early morning is due to a combination of wet scavenging and less convection overnight. The above analysis

suggests that unlike regional NPF in the BL that mostly occurs around noontime, in the FT over tropical oceans,

strong radiation is not a necessary condition for NPF and a large fraction of the background NPF occurs under very

low CS condition in the early morning despite low radiation and actinic flux. It is worth noting that nighttime NPF

has been reported in conditions of low condensation sinks in the upper FT (Lee et al., 2008). However, the

mechanism of nocturnal NPF is not well understood. Given the absence of nighttime measurements during the

campaign, we cannot exclude the possibility that some of the new particles observed in the early morning were

formed during the nighttime.


**4.2 NPF in biomass burning influenced air mass**

Biomass burning is one of the major aerosol sources, emitting not only a large amount of primary particles but also

precursors such as $SO_2$ (Crutzen and Andreae, 1990), DMS (Meinardi et al., 2003), and volatile organic compounds

(VOCs) that lead to secondary aerosol formation (Hennigan et al., 2012; Spracklen et al., 2011; Fiedler et al., 2011;

Meinardi et al., 2003). Few direct measurements of NPF in biomass burning plumes have been reported (Shang et

al., 2018; Vakkari et al., 2018; Hodshire et al., 2021), probably because strong primary particle emission typically

leads to large CS in biomass burning plumes that inhibits NPF. Biomass burning smoke originating from the Borneo

region was sampled during the research flight on 15 September (RF9), during which high $N_{3-10 \, nm}$ was observed

together with a strongly enhanced CO mixing ratio (i.e., 3-5 times above the typical values in background or urban-

influenced air masses). The NPF events in BB-influenced air masses took place at altitudes of ~6.7 km. Five-day

back-trajectories of air masses arriving at different sampling altitudes of RF9 were simulated using HYSPLIT

model. Within the BL, the prevailing wind was from the southwest and the sampled air masses originated from

Borneo regions, where strong biomass burning activities were reported. In contrast, air masses arriving at 6.7 km

came from the west Pacific with no direct influence by biomass burning (Fig. S6). Therefore, the BB-influenced air

mass observed in the FT during RF9 is due to the vertical lifting and detrainment of the biomass burning plume

from BL by convective clouds, instead of direct long-range transport inside the FT from Borneo. The biomass

burning plume had travelled inside the BL across the Sulu Sea from the Borneo (Fig. S6), consistent with previous

findings that transport of smoke to the region mostly occurred within the BL due to strong wind shear during the

southwest monsoon season (Hilario et al., 2020; Xian et al., 2013).

To investigate the potential impact of biomass burning emissions on NPF, we statistically compare NPF observed

during RF9 to background NPF from other flights within the same altitude range (i.e., to account for the influence of

temperature). Because no measurements of non-methane hydrocarbons are available during CAMP²Ex, we use CO

as a surrogate for VOCs emitted from biomass burning. As UV irradiance plays an important role in NPF, the key

variables including $N_{>3\,nm}/N_{>10\,nm}$ and $N_{3-10\,nm}$ during both BB-influenced and background NPF events are compared for the same UV irradiance levels (Fig. 5) such that the role of precursors can be clearly differentiated from other factors.

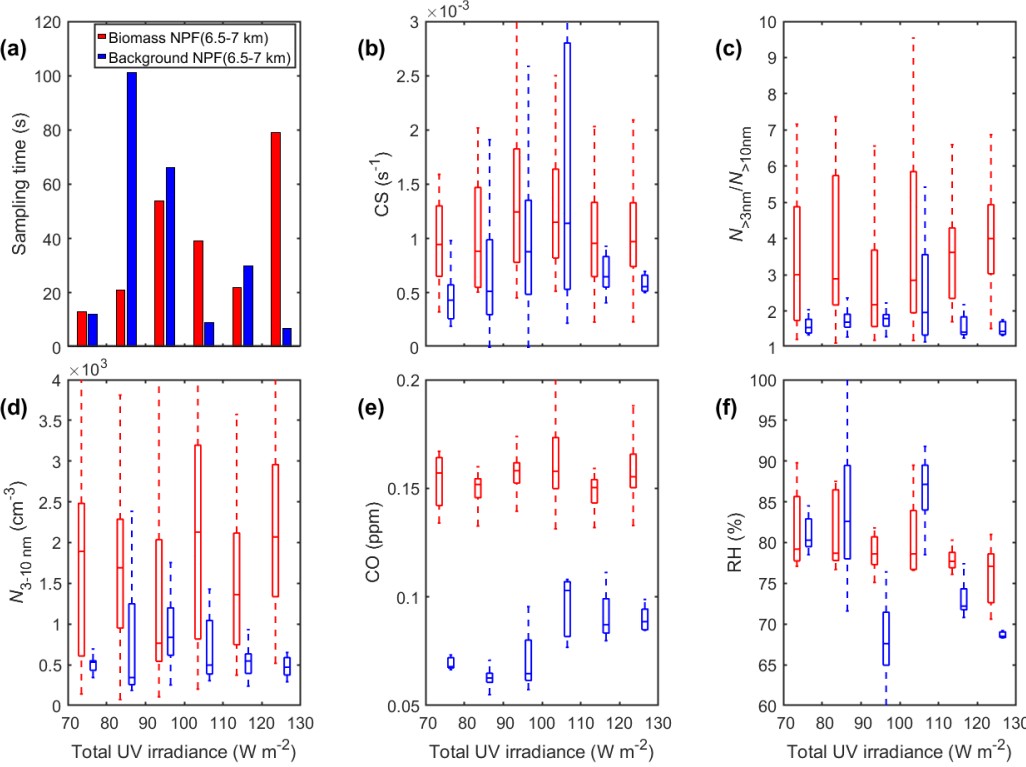

**Figure 5. Comparison between NPF influenced by biomass burning smoke and NPF in background. (a) Sampling time, (b) CS, (c) $N_{>3\,nm}/N_{>10\,nm}$, (d) $N_{3-10\,nm}$, (e) CO and (f) RH are plotted as a function of UV irradiance. The statistics are shown for 10 Wm$^{-2}$ UV irradiance bins (i.e., 70-80 Wm$^{-2}$, 70-80 Wm$^{-2}$, etc.).**

We focus on the comparison for UV irradiance ranging from 70-130 W m$^{-2}$ such that the amount of data for both NPF types are comparable. There exists a substantial fraction of BB-influenced NPF with UV irradiance higher than 130 W m$^{-2}$, whereas few background NPF events at the same altitude range have UV irradiance above 130 W m$^{-2}$. At the same UV irradiance level, BB-influenced NPF occurred with similar or slightly higher CS compared to the background NPF (Fig. 5b), but with much stronger intensity as indicated by higher $N_{>3\,nm}/N_{>10\,nm}$ and $N_{3-10\,nm}$ (Fig. 5c, 5d). This shows that precursors from biomass burning, as indicated by elevated CO mixing ratio (Fig. 5e), strongly enhance NPF. The low CS (i.e., mostly below 0.002 s$^{-1}$, Fig. 5b) and high RH (Fig. 5f) during BB-influenced NPF events suggests that existing particles were efficiently removed through wet scavenging as the biomass burning plume was lifted into the FT by convective clouds. Despite high concentrations of precursors, an efficient removal of existing particles appears to be a necessary condition for NPF to occur in the BB-influenced air masses. Newly formed particles were absent in BB-influenced air masses sampled at the same altitude of 6.7 km during RF9 where the concentrations of non-volatile particles and accumulation mode particles were three times as

high as those during the NPF events. The case presented above shows that convective clouds can efficiently remove existing aerosol in biomass burning plume, leading to low CS similar to that in the background air. The elevated

precursor concentrations in detrained biomass burning plume strongly enhances NPF under the conditions of low CS and sufficient radiation. It remains unclear which nucleation pathway dominates the NPF observed in BB-influenced air masses. Biomass burning emissions include $SO_2$ (Yokelson et al., 2009), ammonia (Hegg et al., 1988), and VOCs such as biogenic VOCs (e.g., terpenoids), aromatics, and heterocyclic compounds (e.g., furan) (Ahern et al., 2019; Akherati et al., 2020; Gilman et al., 2015; Yee et al., 2013). The oxidation of $SO_2$ and VOCs can produce

sulfuric acid and highly oxygenated organic molecules (HOMs), and the mixtures of sulfuric acid, ammonia and HOMs have been shown leading to strong NPF (Lehtipalo et al., 2018). More measurements, including the precursors and nucleating species, are required to understand nucleation mechanisms in BB-influenced air masses.

**4.3 NPF influenced by urban emissions**

Besides background and BB-influenced NPF, NPF events were also observed in urban-influenced air masses during

CAMP[2]Ex. These urban-influenced NPF events occurred under very different conditions, (e.g., RH, CS) and are classified into different k-means clusters. Therefore, the discussion of the urban-influenced NPF events will follow the classification by k-means clustering (Fig. 3). A large fraction of cluster #1 is classified as urban-influenced, which is mostly from RF7 and RF8 at an altitude of 5.5-6.5 km, while a small fraction of cluster #3 and the majority of cluster #4 represent urban-influenced NPF at altitudes above 7 km. The remainders are distributed throughout

cluster #5-6 and occurred with elevated CS (Fig. 3c). The effects of urban emissions on NPF depend on the age of the urban plume and altitudes, as detailed below.

**4.3.1 NPF over coastal regions and land at altitudes of 5.5-6.5 km**

Urban-influenced NPF events classified into cluster #1 share some similar features with the NPF observed in BB-influenced air masses. The locations of these urban-influenced NPF events are shown in Fig. S7. Measurements on

13 September 2019 show elevated $N_{>3\ nm}/N_{>10\ nm}$ and $N_{3-10\ nm}$ during the level flight at ~5.8 km near Manila. In addition, on 8 September 2019, extremely high $N_{>3\ nm}/N_{>10\ nm}$ values up to 40 and $N_{3-10\ nm}$ above 4000 $cm^{-3}$ were observed at altitudes of ~6.3 km over the West Pacific about 50 km away from the coastline. These events together represent over 70% of cluster #1, and the general features include low CS, high UV irradiance, and high RH, similar to BB-influenced NPF events.

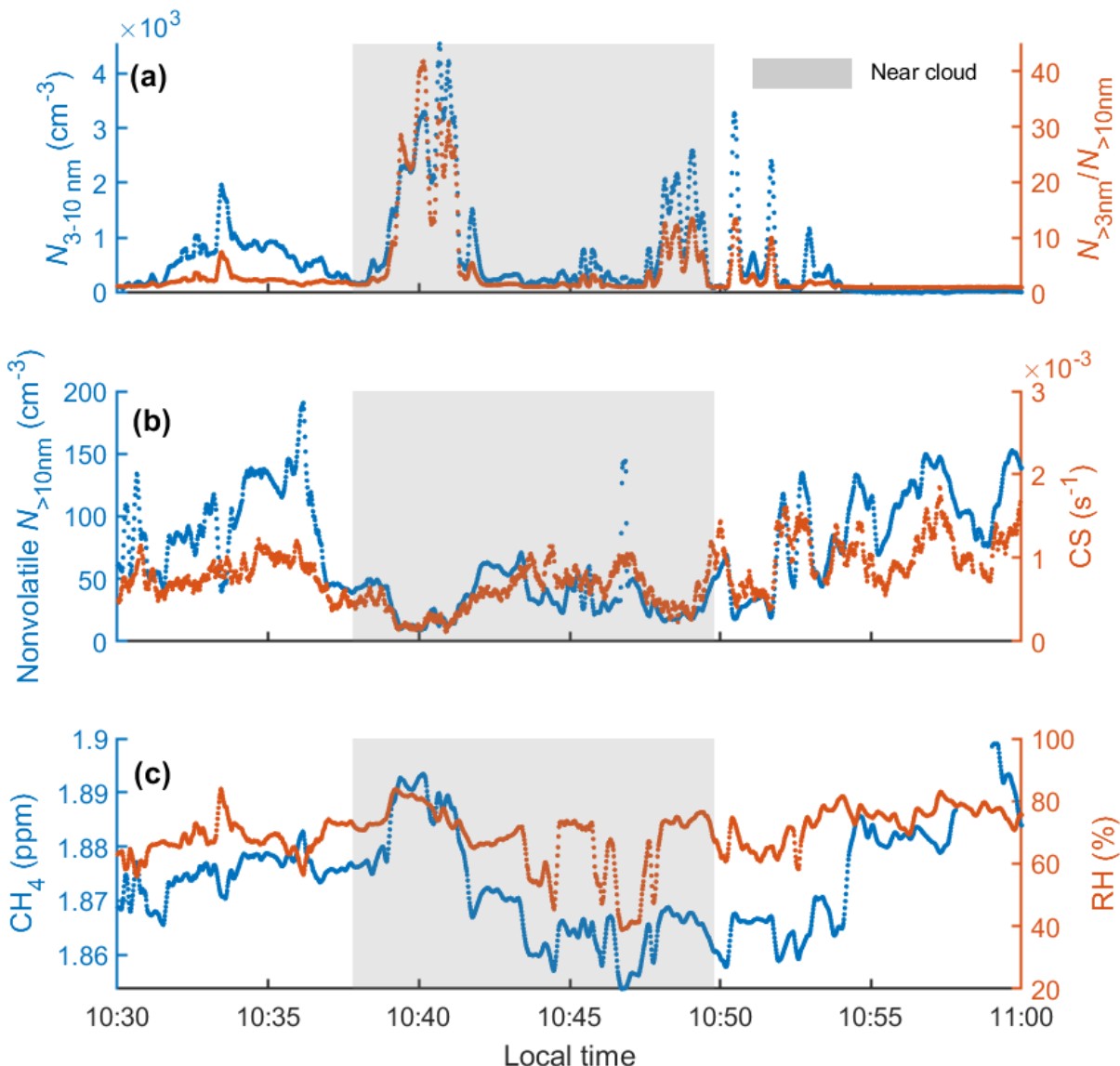

**Figure 6. Time series plots of a level flight segment at 6.5 km with NPF observed in urban-influenced air masses near cumulus clouds during RF7 (8 September 2019), including (a) $N_{3-10\ nm}$ and $N_{>3\ nm}/N_{>10\ nm}$, (b) non-volatile $N_{>10\ nm}$ and CS, and (c) $CH_4$ mixing ratio and RH.**

Figure 6 shows the key variables during a representative urban-influenced NPF event in cluster #1, which was observed at 6.5 km over the ocean east of Luzon during RF7 on 8 September 2019. The ratio $N_{>3\ nm}/N_{>10\ nm}$ drastically increases around 10:40 near convective clouds (based on the video from the forward-looking camera). At the same time, both the CS and concentration of non-volatile particles (nonvolatile $N_{>10\ nm}$) decrease while RH and $CH_4$ concentration become elevated, indicating the uplift of humid and urban-influenced air from lower altitudes and efficient removal of pre-existing particles. The concurrence of drastically increased $N_{>3\ nm}/N_{>10\ nm}$ and elevated $CH_4$

suggests urban emissions contribute to the production of nucleation species and NPF. Here $CH_4$ is used as a surrogate for urban emitted precursors, which typically include $SO_2$, gaseous sulfuric acid, and organic species (Zhang et al., 2012). A similar positive correlation between $N_{>3\,nm}/N_{>10\,nm}$ and $CH_4$ concentration is also evident during the leveled box flight segment of RF8, which took place close to Manila (not shown). Compared to most other NPF events, these events were observed closer to urban areas over the land and therefore more likely

influenced by fresh urban emissions. The contribution of urban emitted precursors to the NPF is also supported by statistical comparisons of $CH_4$ concentration, UV irradiance, and RH between such NPF events and non-NPF periods at the same altitude (5.5-6.5 km) and time of day as a function of CS (Fig. S8 and Supplementary Section 3). When CS is below 0.0015 $s^{-1}$, urban-influenced NPF has a similar level of UV irradiance but higher $CH_4$ concentration and RH compared to the non-NPF periods, suggesting that precursors freshly emitted from urban

areas, including $SO_2$, gaseous sulfuric acid, and VOCs, likely contribute to the production of nucleation species and NPF.

**4.3.2 NPF influenced by aged urban plumes**

Part of cluster #3 and cluster #4 represent urban-influenced NPF observed at higher altitudes (~7-8.1 km) than urban-influenced NPF observed at 5.5-6.5 km (i.e., cluster #1). In addition to the difference in altitude, these NPF

events took place over the open ocean and some of them exhibit relatively high CS (e.g., cluster #3, Fig. 3c). Figure 7 shows representative examples of such NPF events, which were observed at an altitude of 7.1 km during RF10 over the west Pacific, 600 km away from the coast. During the NPF events, $N_{>3\,nm}/N_{>10\,nm}$ and $N_{3-10\,nm}$ reach 3 and 2000 $cm^{-3}$, respectively. The elevated $N_{>3\,nm}/N_{>10\,nm}$ and $N_{3-10\,nm}$ coincide with elevated RH and reduced CS, and both $N_{>3\,nm}/N_{>10\,nm}$ and RH are anti-correlated with $CH_4$ concentration during the period (11:25-11:32, local time),

indicating particle formation in cloud outflow regions with reduced $CH_4$ concentration. Simulated back-trajectories suggest the air masses with elevated $CH_4$ level (i.e., around 1.9 ppm) at ~7.1 km was influenced by aged urban plume transported from East Asia. The anti-correlations between RH and $CH_4$ indicate that humid background air (i.e., with low $CH_4$ concentration) was lifted by convective clouds and mixed into the aged urban plume. We expect the reactive precursors in the aged urban plume are mostly consumed during the long-range transport, while $CH_4$

concentration and CS remain relatively high due to longer lifetimes in the FT. As a result, NPF only occurs when the aged plume is mixed with sufficient air detrained from convective clouds, which is expected to have reduced CS and elevated concentration of reactive gases such as DMS. Therefore, the aged urban plume tends to suppress NPF instead of promoting it as in the air masses influenced by fresh urban emissions shown in Sect. 4.3.1.

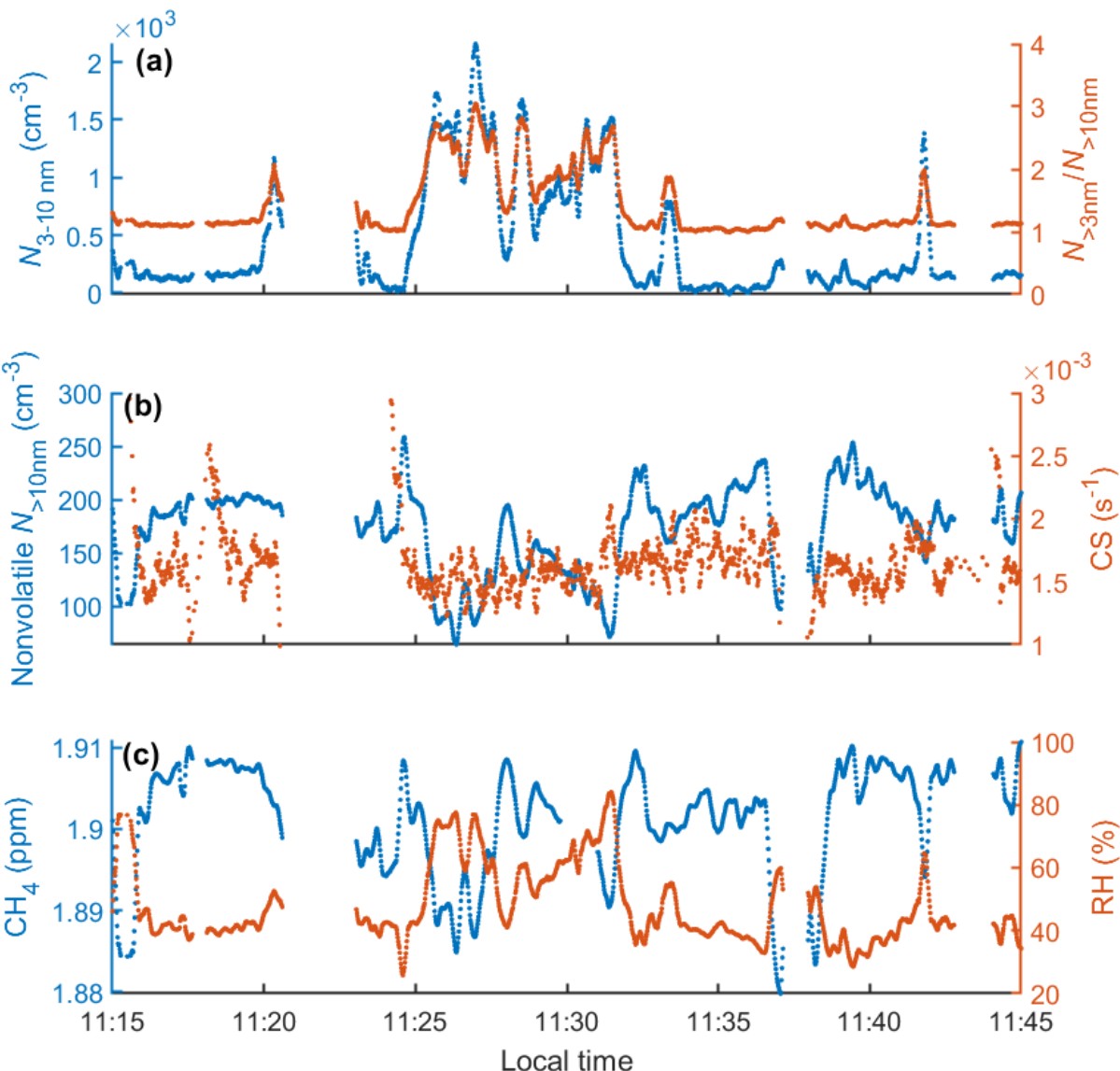

**Figure 7. Time series plots of a level flight segment at 7.1 km during RF10 (16 September 2019) where NPF was observed, including (a) $N_{3\text{-}10\,\text{nm}}$ and $N_{>3\,\text{nm}}/N_{>10\,\text{nm}}$, (b) non-volatile $N_{>10\,\text{nm}}$ and CS, and (c) CH$_4$ mixing ratio and RH.**

### 4.3.3 Urban influenced NPF with high CS

Unlike most NPF events during CAMP²Ex, a small fraction of urban-influenced NPF events occurred under high CS. These events were grouped into k-means clusters #5 and #6. Figure S9 shows increasing $N_{3\text{-}10\,\text{nm}}$ with concentration of accumulation mode particles (i.e., $N_{>100\,\text{nm}}$) during an example of such NPF events. This example was observed at 4.8 km over Metro Manila during RF18, which was designed to sample urban plumes from Metro

Manila. As new particles typically form under low CS conditions, a negative correlation between $N_{3\text{-}10\,nm}$ and $N_{>100\,nm}$ is expected. The positive correlation, together with the sampling location, raise the possibility that both $N_{3\text{-}10\,nm}$

and the accumulation mode particles might originate from primary emissions in Metro Manila. Previous studies show that aerosol particles with diameters of a few nanometers can form as the fresh exhaust from diesel/gasoline engines rapidly cools. While these nanoparticles are formed through nucleation, they are often considered "primary" as the nucleation process occurs very close to the sources (Uhrner et al., 2011; Wehner et al., 2009). If the elevated $N_{3\text{-}10\,nm}$ was indeed due to primary emissions in Metro Manila, we would expect even higher $N_{3\text{-}10\,nm}$ at lower

altitudes. However, no NPF events were identified when P-3B sampled in the metro Manila regions below 4.8 km. In addition, albeit from a different flight, the vertical profiles of aerosol and trace gases during a descending leg over Lingayen Gulf (RF8, Fig. S10) show that the small particles with diameters between 3 and 10 nm are secondary despite a positive correlation between $N_{3\text{-}10\,nm}$ and $N_{>100\,nm}$. The vertical profiles show several detrainment layers with elevated $N_{3\text{-}10\,nm}$ from 2.5 km up to 4.5 km, whereas the small particles are mostly absent below 2.5 km. The

comparisons among the different layers show that $N_{3\text{-}10\,nm}$ increases while $N_{>100\,nm}$, CS, and CO decrease with altitude, indicating the observed small particles formed in the detrainment layers instead of originating from the primary emissions near the surface. The mechanism for this type of NPF is likely similar to those observed in polluted urban BL (Alam et al., 2003; Zhu et al., 2014; Yao et al., 2018), where high concentrations of precursors make nucleation and particle formation possible despite the high CS. The absence of NPF below 2 km may result

from a combination of higher condensation sink and warmer temperature compared to those in the detrainment layers at higher altitudes. Earlier studies show competing effects of temperature on NPF and particle growth (Stolzenburg et al., 2018; Ye et al., 2019). While a higher temperature leads to higher reaction rate and concentration of HOMs, it also strongly increases the volatility of organic species, therefore slowing down or even inhibiting NPF and early growth of newly formed particles.

**4.3.4 Discussion – impact of urban emissions on NPF in tropical FT**

Impact of urban emissions on NPF in the tropical marine FT is poorly understood as previous measurements were mostly carried out under pristine conditions. The analyses presented above show that depending on the age and altitude, urban emissions can have different effects on NPF. Above 5.5 km (i.e., approximately the freezing level), convectively detrained fresh urban plumes have low CS due to efficient wet scavenging of existing particles, and

elevated precursor concentrations contribute to and enhance NPF under the low CS condition in the outflow regions. At lower altitudes (i.e., below freezing level), NPF takes place in detrained fresh urban plumes with higher CS compared to the background. The higher CS is likely due to less efficient wet scavenging at these lower altitudes. High concentrations of precursors from the fresh urban emissions likely made these NPF possible despite relatively high CS. The species participating in these NPF events may include sulfuric acid and amines (Yao et al., 2018).

Future measurements, including the precursors and nucleating species, are needed to elucidate the nucleation mechanisms in the air masses influenced by fresh urban emissions. In aged urban plumes over 7 km, reactive precursors are mostly consumed during the long-range transport from East Asia, while CS remains relatively high.

As a result, the aged urban plumes tend to inhibit NPF instead of promoting it as in the case of fresh urban emissions.

## 5 Summary and conclusions

In this study, we examine NPF in the tropical marine FT in the altitude range of 3-8.5 km using airborne measurements collected during the CAMP$^2$Ex campaign. NPF events were classified based on air mass types, including background, biomass burning-influenced, and urban-influenced. The features of key variables, including RH, CS, UV irradiance as well as concentrations of trace gases are presented for different NPF types and over different altitude ranges. No newly formed particles were observed below 3 km, and NPF was rare and mostly observed in urban-influenced air between 3 and 5.5 km. Vast majority of the NPF events were observed above 5.5 km in air that was processed by convective clouds and with low CS. The frequency of NPF increases with altitude, reaching about 50% at 8 km. There is a drastic decrease in NPF frequency from the southwest monsoon to the monsoon transition period, which is attributed to the increased CS resulting from decreased convective activity (i.e., less efficient removal of existing particles) and more frequent transport of aged urban pollution associated with altered meteorological conditions.

Two different types of NPF in background air were observed in the vicinity of convective clouds. One type was observed under the condition of strong UV irradiance around noontime as in previous studies. In contrast, the second type occurred in the early morning with some of lowest CS observed during CAMP$^2$Ex. The very low CS is attributed to a combination of wet scavenging and less convection (i.e., reduced vertical transport of aerosol particles from near surface to the FT) over night, and it likely makes the second type of background NPF possible despite low UV irradiance and actinic flux.

NPF was observed in BB-influenced air at altitudes of ~ 6.7 km. Convectively detrained biomass burning plume enhances NPF because of elevated precursor concentrations and efficient scavenging of pre-existing particles. The effect of urban emissions on NPF depends on the age of the urban plume and altitude. Newly formed particles in air masses influenced by fresh urban emissions were observed under the condition of low CS in the outflow regions at altitudes between 5.5 and 6.5 km. The NPF was promoted by elevated concentrations of precursors from the fresh urban emissions. At lower altitudes (i.e., below freezing level), a small number of NPF events were observed in detrained fresh urban plumes with higher CS compared to the background. High concentrations of precursors from the fresh urban emissions likely made these NPF possible despite relatively high CS. Above 7 km, NPF was observed when the background humid air was lifted by convective clouds and mixed into the aged urban plumes. The reactive precursors in the aged urban plumes are mostly consumed during the long-range transport from East Asia, while CS remains relatively high. As a result, the aged urban plumes inhibit NPF instead of promoting it as is the case for the fresh urban emissions.

This study highlights the competing influences of different variables and interactions among anthropogenic emissions, convective clouds, and meteorology, which lead to NPF under a variety of conditions and altitudes. Most

earlier studies found that the NPF typically occurs under the conditions of strong solar radiation around noontime. Here we show NPF can occur in the FT with very low CS in the early morning, despite the low actinic flux. Depending on their age, urban emissions can either enhance or inhibit NPF in the tropical marine FT. Biomass

burning plumes strongly enhance NPF in the outflow region of convective clouds once existing particles are efficiently removed by wet scavenging. Due to the lack of measurements of precursors, the nucleation pathways of NPF in different air mass types remain unclear and should be examined in future studies. The impact of urban and biomass burning emissions on NPF, and the subsequent formation of CCN will also need to be examined in the future by combining field observations and model simulations.


*Data Availability.* CAMP$^2$Ex observational datasets are available at https://asdc.larc.nasa.gov/project/CAMP2Ex. HYSPLIT data are accessible through the NOAA READY website (http://www.ready.noaa.gov). The code used to generate the figures is available upon request.

*Author Contributions.* JW and QX designed the study. JZ, YW, LZ, EC, EW, CR, JD, GD, KS, SW, SS, and PL
carried out the measurements and data reduction. QX and JW led the data analysis and the preparation of manuscript, with contributions from all authors. We thank Michael Jones and Adam Bell for their comments on the manuscript.

*Competing interests.* One of the co-authors is a member of the editorial board of *Atmospheric Chemistry and Physics*. The peer-review process was guided by an independent editor, and the authors have also no other
competing interests to declare.

*Acknowledgement.* We acknowledge the funding support from National Aeronautics and Space Administration (grant no. 80NSSC19K0618).

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
