# Peer review of "New Particle Formation in the Tropical Free Troposphere during CAMP2Ex: Statistics and Impact of Emission Sources, Convective Activity, and Synoptic Condition"

_Atmospheric Chemistry and Physics, 2022_

## Referee Comment (RC2)

Review on 'New Particle Formation in the Tropical Free Troposphere during CAMP2Ex: Statistics and Impact of Emission Sources, Convective activity, and Synoptic Condition' by Qian Xiao et al.

General Comments

The submitted publication presents aircraft measurements with the focus on new particle formation events. Data were obtained during a field campaign in the Tropics and a very complex set of data is available. Data are observed using different methods with regard to e.g., air mass characteristics and cluster analysis. This is a valuable work and should be published. But the presented manuscript needs some revision and clarification. Furthermore, my question below, if the thermodenuder was in use, needs to be answered. If the TD was not heated and the comments below are considered, I support a publication in ACP.

There is a really basic question for me, rising up in the description of instruments. From line 115 on it is explained that two particle counters are measuring number concentrations above 3 nm and 10 nm. In the following table 1, all instruments are listed and there is an addition for the second CPC, that it measures downstream of a thermodenuder and thus, measures the number concentration of non-volatile particles above 10 nm. In my view, this is a significant difference, if $N_{10}$ is the total number concentration > 10 nm or the number concentration of non- volatile particles > 10 nm. This difference is crucial for all conclusions coming from this paper and thus, it is difficult to formulate a review without knowing what is investigated here. I simply assume, that the thermodenuder is not in use, i.e. not heated, otherwise the data don't make sense to me. If this is not the case and the thermodenuder was heated the interpretation has to be rewritten because it is a different parameter. I went through the individual section in more detail below.

Section 3 Overall Statistical Analysis

This section should be structured in a clearer way. I do not think that not all subsections fit under 'statistical analysis', e.g., the dependence of Monsoon transition. Here, another headline would help. My question is: are all subsections really needed? What are the main conclusions? The figures are not easy to understand, since every figure is slightly different.

There are some sentences, which are difficult to digest, e.g. (l. 201f.), 'Most NPF events were observed above 3 km when RH exceeded 50%, and only about 2% of total NPF was observed at 3.5-5 km.' From this, I would conclude that most NPF events occur above 5 km?! Such sentences should be homogenized.

Figure 2a) it is hard to distinguish low NPF frequencies from zero. At the lowest point, it looks like the frequency in background air is significantly above zero, but it is stated above: 'Figure 2a shows that below 5.5 km, no NPF events were observed in background or BB-influenced air masses (l. 224)'

This does not fit for me; such results have to be clear and comprehensible. Please check this para

Does Figure 2 include all data?

Figure 3 includes the same data, but just divided into the two periods? There are too many similar looking pictures, which are not so easy to interpret.

Some words summarizing the main results would really be helpful in my view.

Section 4: Characteristics of NPF in Different Air Mass Types

In subsection 4.1 NPF events in Background air were analyzed. Here the two different clusters were shown in the following. In the subsection about biomass burning those events from a certain height region were compared with background cases. In the third subsection, the urban NPF cases were compared with non-NPF events. This is really confusing. Here, a more homogeneous way should be selected and maybe one figure containing all three airmasses, maybe in comparison with non-NPF cases can replace some of the figures. There are definitely too many and too different figures in the manuscript and this should be changed.

Also, the mix of statistical analysis, cases studies as time series and profiles is confusing and should be better structured/motivated.

Summary:

This section summarizes the most important finding of the study. However, I miss a bit more interpretation. Furthermore, it should be estimated how relevant such NPF processes are at all. Here it should be stated, which type of NPF plays a significant role and which is more a minor process and so on. In general, how relevant is the free tropospheric NPF for this region, for other regions? Is it possible to conclude such a statement from these results?

In the summary the term condensation sink is used while in figures always surface area is shown. Of course, these parameters are similar, but not identical. I would prefer if CS would be used throughout the whole paper.

Maybe some more comparison of the different air mass types is possible, one main difference is probably the condensation sink, but could the authors speculate also about differences in precursor gases? What about dynamics? Local mixing processes can foster nucleation and growth, is this relevant here?

---

## Author Comment (AC1)

**Response to RC1**

We thank the reviewers for their thoughtful and constructive comments. Please find below the response to each comment or question, including notations of improvements to the manuscript. Reviewer comments are in blue fonts and responses are in black. Changes to the manuscript are highlighted in italics.

- One general concern, which needs to be at least thoroughly discussed in the manuscript, is that the authors often connect the observation of sub-10 nm particles (elevated $N_{>3nm}/N_{>10nm}$ ratio) directly to the observed external conditions and link this occurrence of "NPF" to them. However, in the FT new particle growth rates might be just in the order of <1 nm h$^{-1}$, which means that the observation of 3-10 nm particles could originate from particle formation processes which already occur over several hours (up 15-20 hours, if we assume GR=0.5 nm h$^{-1}$ and the growing nucleation mode to be located at around 8-10 nm, which is not resolved by the simple $N_{>3nm}/N_{>10nm}$ ratio due to missing size-distribution information). In that case, the current conditions under which the sub-10 nm particles are observed might not at all representative for the conditions under which the new particles might have been forming. The authors should discuss potential implications of this within their analysis.

**Response:** This is a very good point. Ideally, the concentration of incipient particles (i.e., particles with diameter around 1.5 nm) should be used to identify new particle formation (NPF) events. However, given the challenges of measuring particle concentration below 2 nm onboard research aircraft, many airborne studies have used the ratio of $N_{>3nm}/N_{>10nm}$ (e.g., Zheng et al., 2021) and/or $N_{3-10\ nm}$ (e.g., Crumeyrolle et al., 2010) to characterize NPF events.

As the reviewer pointed out, the observation of 3-10 nm particles likely occurs several hours after the particle formation. As incipient particles are efficiently removed by coagulation inside clouds, we expect the air mass with elevated $N_{>3nm}/N_{>10nm}$ remained cloud free and did not experience precipitation since the recent particle formation. Therefore, condensation sink (CS) and relative humidity (RH), which are among the NPF related parameters examined in this study, are unlikely to vary drastically over a period of several hours following the particle formation. New particle formation and subsequent particle growth can lead to an increase of CS. For elevated $N_{>3nm}/N_{>10nm}$ observed under conditions of low CS, the formation of new particles likely had occurred with comparable or even lower CS. UV irradiance has a strong diurnal variation and depends on the cloud condition, and it can change substantially over a period of several hours. In this study, most NPF events (i.e., elevated $N_{>3nm}/N_{>10nm}$) were observed at noontime under higher levels of UV irradiance compared to the non-NPF periods at the same altitude, consistent with earlier studies showing that solar radiation was generally higher during NPF event days compared with non-event days. Some of NPF events were observed under conditions of low UV irradiance, and the potential mechanisms are discussed in Section 4.1.

Following the suggestions, we have included the discussion below in section 3.3:

*Because it takes some time for incipient particles to grow into the 3-10 nm size range, the NPF events identified here using $N_{>3nm}/N_{>10nm}$ value are likely several hours after the formation of the new incipient particles, depending on the actual growth rate. As the incipient particles are*

*efficiently removed by coagulation inside clouds, we expect that air masses with elevated $N_{>3nm}/N_{>10nm}$ remained cloud free and did not experience precipitation since the recent particle formation. Therefore, CS and RH, which are among the NPF related parameters examined in this study, are unlikely to vary drastically over a period of several hours following the particle formation. New particle formation and subsequent particle growth can lead to an increase of CS. For elevated $N_{>3nm}/N_{>10nm}$ observed under conditions of low CS, the formation of new particles likely have occurred with comparable or even lower CS. UV irradiance has a strong diurnal variation and depends on the cloud condition, and it can change substantially over a period of several hours. In this study, most NPF events (i.e., elevated $N_{>3nm}/N_{>10nm}$) were observed at noontime under higher levels of UV irradiance compared to the non-NPF periods at the same altitude, consistent with earlier studies (Kerminen et al., 2018) showing that solar radiation was generally higher than during NPF event days compared with non-event days. Some of NPF events were observed under conditions of low UV irradiance, and the potential mechanisms are discussed in Section 4.1.*

- As this study focusses on NPF it would be very useful to use the quantity of condensation sink (CS in $s^{-1}$, as defined by Dal Maso et al. 2015) instead of surface area. If the CS is even calculated using assumptions on the hygroscopicity of FT particles, it would much better represent the actual sink of condensable vapors and, related to it, the coagulation sink of newly formed clusters. This would incorporate the RH information into that parameter (high RH means also even higher sink typically at the same dry aerosol surface area. Related to that: Do FIMS and LAS actually dry the sample? I am missing that information from the Methods section). This would significantly help to relate the observations of sub-10 nm particles in the FT to NPF at many other locations where CS is reported. Related to comment 1) a comparison of CS to a potential GR then also directly gives an approximation of the survival probability.

**Response:** We thank the reviewer for this suggestion. The FIMS and LAS measured dry aerosol size distribution. We derived an average hygroscopicity parameter (κ) using AMS measurements for aerosols sampled above 5 km, where the vast majority of the NPF events was identified. The CS was derived from the ambient aerosol size distribution, which was calculated from the average aerosol hygroscopicity, ambient RH, and measured dry aerosol size distribution. The aerosol surface area was replaced by CS in all relevant figures and discussions were modified accordingly throughout the main text. We reworked the line 133-136 as follows to describe the approach of calculating CS:

*Condensation sink (CS) reflects how quickly condensable vapors will condense on the existing aerosol (Dal Maso et al., 2002). We calculated CS from the ambient aerosol size distribution (Kulmala et al. 2012), which was derived by combining dry particle size distribution measured by FIMS (10-600 nm) and LAS (600-1000 nm), ambient RH, and an average hygroscopicity parameter (κ). Aerosol mass spectrometer (AMS) measurements show that on average, $(NH_4)_2SO_4$ represents 90% of the $PM_1$ mass above 5 km, where the vast majority of the NPF events was identified. A κ value of 0.53 was therefore applied to calculate particle hygroscopic growth factor at ambient RH (Petters and Kreidenweis, 2007) for each size bin of the combined dry size distribution.*

We also included the comparison of CS to previously reported values in section 3.2, where Fig. 2 presents the vertical profile of CS for NPF and non-NPF periods of the whole mission:

*Figure 2 shows that most NPF events occur when CS is below 0.002 s$^{-1}$. For NPF events observed above 6.5 km, the median CS value is mostly below ~ 0.001 s$^{-1}$, comparable to the CS below 8 × 10$^{-4}$ s$^{-1}$ globally in the tropical mid-FT reported by Williamson et al. (2019).*

[Figure]

***Figure 2.** (a) The vertical profile of NPF frequency for the three air mass types. NPF Frequency is defined as the ratio of total duration of NPF period to the total sampling time outside of the clouds for each airmass type. Also shown are the comparison of (b) relative humidity (RH) and (c) condensation sink (CS) between NPF and non-NPF periods, where black denotes non-NPF and orange denotes NPF.*

As the reviewer points out, the survival probability depends on both growth rate and coagulation sink (Kulmala et al., 2012; Kerminen et al., 2018; Kuang et al., 2009; Westervelt et al., 2013). However, we cannot derive the particle growth rate from the measured aerosol size distribution because the CAMP$^2$Ex flights were not designed to track the aerosol evolution in the same air masses (i.e., Lagrangian sampling).

- I am missing the clarity in the analysis, especially from Section 4 onwards, where I find many Figures difficult to read (and potentially unnecessary for the main text), while other Figures would have been helpful. I have some major points here:
  My biggest scientific concern related to that is that the authors often compare conditions of NPF of a certain cluster and/or attributed air mass origin with conditions of no-NPF at the same altitude. However, as altitude is not the only variable which leads to the classification of a certain NPF event in a certain cluster and/or attributed air mass origin, this comparison is in my opinion not very useful. I thus suggest leaving the no-NPF periods from Fig. 4b-f and delete Fig. 9 (I think Fig. 8 already shows the most important conclusion for Section 4.3.1).

Figure 4 (now as Fig. 3) has been modified as suggested (see modified Figure below). All the non-NPF data have been removed and panel c now shows CS.

[Figure]

*Figure 3. (a) Amount of data classified into each cluster and contributions of different air mass types. The other five panels compare the clusters in terms of (b) total UV irradiance, (c) CS, (d) the ratio of number concentration of particle larger than 3 nm to that of particle larger than 10 nm ($N_{>3\,nm}/N_{>10\,nm}$), (e) number concentration of particles between 3 and 10 nm ($N_{3-10\,nm}$) and (f) RH.*

Fig. 9 supports the conclusion of Section 4.3.1 from a statistical perspective. We have moved Fig. 9 to SI.

- 6 and Fig. 11, Fig. 12 are in my opinion supportive material but no major results and could easily be moved to the SI. I also consider Fig. 1 not as the most important Figure to start the manuscript with. It remains unclear what is meant by "sampling data count". Why not color the bar plot with the amount of background, biomass-burning and urban emission related NPF and no-NPF observed during the flight and add it as an additional panel to Figure 2.

Following the reviewer's suggestion, we have moved Figures 6, 11 and 12 to the SI. Regarding Fig. 1, sampling data count refers to the total amount of 1-second data collected during each flight. Figure 1 has been revised accordingly. However, including Fig. 1 as an additional panel of Fig. 2 will make Fig. 2 very crowded. Therefore, Figure 1 is now moved to SI instead. The revised Fig. S1 is shown below, with air mass types colored the same way as in Fig. 3.

[Figure]

*Figure S1. (a) NPF frequency for each flight, i.e., the ratio of NPF events to the total amount of data for each RF. (b) Bar chart of the amount of data sampled during each RF and color coded by air mass types.*

- To put all the flights and observations into a clearer relation I suggest making an additional version of Figure S1 (maybe a second panel), where all flights are represented as black lines when no NPF is observed and colored when NPF is observed with the same coloring as in Fig. 4a (blue for background, yellow for urban and red for biomass-burning). Moreover, please add a legend to the already existing panel relating the current colors to the number of the RF. Please also improve the resolution of that Figure significantly to make it better readable. If all the updates are made according to my suggestions, Fig. S1 could even move to the main MS as it gives the reader a quick geographical overview. Please also indicate the position of Manila on the map!

This is an excellent suggestion. Figure S1 has been modified as suggested and moved to the main text as Fig. 1 (shown below).

[Figure]

**Figure 1.** *(a) NPF events during the whole mission color coded by three air mass types (background, BB-influenced and urban-influenced). (b) Locations of 19 flight tracks.*

Section 3.1 now reads:

*There was a total of 19 research flights (RFs) during CAMP²Ex.* Figure 1 shows an overview of the flight tracks and the locations where NPF in three major air mass types were observed. *These RFs covered the ocean east and west of Luzon Island, and two of them (RF8 and RF18) sampled over Luzon Island and upwind/downwind of Metro Manila. The date and sampling areas of all RFs, together with the duration and key variables of observed NPF events, are presented in Table S1. Most NPF events were observed above 5 km when RH exceeded 50%. A few short periods with elevated $N_{>3nm}/N_{>10nm}$ (not counted as NPF events) were observed within the boundary layer about 50 kilometers downwind west of metro Manila during RF18, which are closely associated with shipping and/or urban emissions. These NPF events likely occurred immediately following the dilution of vehicle and engine emissions (e.g., Uhrner et al., 2011; Wehner et al., 2009), and they are not included in further analyses. NPF frequency, defined as the ratio of the sampling time when new particles were observed to the total flight time, decreased drastically starting from RF11 on 19 September and no events were observed from RF12 through RF17 as shown in Fig. S1. This sudden decrease in NPF frequency coincided with the early monsoon transition starting on 20 September (Hilario et al., 2021).*

Minor comments

- Page 2, 56-57: Please provide a reference for that statement.

A reference has been added in line 56-57 and the text now reads:

*Essentially all long-term surface measurements show that the average solar radiation intensity is stronger during NPF event days compared with non-event days (Kerminen et al., 2018).*

- Page 3, line 90: "from the perspective of galactic cosmic rays". What is this supposed to mean? GCRs are known to enhance NPF of weakly binding systems such as $H_2SO_4+NH_3$ and HOMs. The work of the CLOUD team (Kirkby et al., 2011 and 2016, Nature) should be mentioned as they have provided the most thorough investigations of the role of GCRs in NPF so far.

Thanks for suggesting the reference. Line 89-91 now reads:

*Kirkby et al. (2011) found that ion-induced binary nucleation associated with galactic cosmic ray can occur in mid-troposphere but is negligible in the boundary layer, while the strongest aerosol formation takes place in upper troposphere over tropic ocean (Kazil et al., 2006).*

- Page 3, line 107: In accordance with the guidelines of ACP, please refrain from citing unpublished references. See guide for authors: "Works cited in a published manuscript should be published already, accepted for publication, or available as a preprint with a DOI".

The paper has been published in BAMS online and reference is now added: *Reid, J. S., and Coauthors, 2023: The coupling between tropical meteorology, aerosol lifecycle, convection, and radiation, during the Cloud, Aerosol and Monsoon Processes Philippines Experiment (CAMP2Ex). Bull. Amer. Meteor. Soc., https://doi.org/10.1175/BAMS-D-21-0285.1, in press.*

- Page 5, line 134-135: "combined size distribution from multiple instruments, including FIMS and LAS". Are FIMS and LAS the only instruments combined here than delete "multiple instruments, including", or were there other instruments incorporated in that combined size distribution than mention them explicitly before. Were the size-distributions just added as the instruments covered different size-ranges or was there some combining instrument inversion applied?

We combined size distributions from FIMS (10-600 nm) and LAS (600-1000 nm). No combined data inversion was applied. We have clarified this in the revised manuscript. Please refer to the response to the 2nd major comment regarding the calculation of CS.

- Page 5, line 149: How is the uncertainty of the ratio defined? By the variance of the data within the 10 second interval, or by some pre-set error on the estimates of $N_{>3nm}$ and $N_{>10nm}$? Please specify.

The uncertainty of the concentration ratio was derived from uncertainties in $N_{>3nm}$ and $N_{>10nm}$ using uncertainty propagation. The uncertainties in $N_{>3nm}$ and $N_{>10nm}$ are calculated based on the counting statistics of the two CPCs. The approach is detailed in Zheng et al. (2021) as referenced in line 143-144.

- Page 6, line 160-161: Such periods are often called "undefined" in typical NPF studies.

We agree. On the other hand, the analysis is focused on identified NPF events in this study. We prefer not to introduce additional definitions that are rarely used in the manuscript.

- Table 2: It could be useful to also give the data ranges for the different mean values and clusters.

Standard deviation has been added to Table 2 (see below for revised Table 2).

**Table 2. General statistics of key parameters for the 6 clusters identified using k-means classification.**

| Cluster # | Number of events | Amount of 1-s data | Mean±std altitude, m | Mean±std temperature, °C | Mean±std UV irradiance, W m$^{-2}$ | Mean±std RH, % | Mean±std CS, $10^{-3}$ s$^{-1}$ |
|---|---|---|---|---|---|---|---|
| 1 | 35 | 5550 | 6104.6±591.9 | -4.6±3.3 | 108.8±13.6 | 75.4±9.0 | 1.1±0.5 |
| 2 | 20 | 3870 | 7708.8±433.2 | -15.2±2.4 | 104.5±13.1 | 79.8±8.5 | 2.0±0.7 |
| 3 | 13 | 3960 | 6392.4±369.8 | -7.3±1.8 | 60.9±14.4 | 82.6±7.0 | 1.1±0.5 |
| 4 | 9 | 1190 | 7532.1±438.2 | -12.9±2.5 | 118.8±21.2 | 33.3±13.5 | 1.2±0.6 |
| 5 | 11 | 790 | 6698.5±650.7 | -10.3±4.2 | 93.7±23.6 | 61.3±6.3 | 5.1±1.2 |
| 6 | 7 | 400 | 3959.3±671.3 | 4.2±4.3 | 58.1±24.1 | 74.2±10.5 | 2.9±2.0 |

- Page 7, line 183-189: In the absence of an accessible version of DiGangi et al. (see above comment) this needs to be more detailed as this is a central part of how the different NPF occurrences have been specified.

Joshua DiGangi suggested that the description of the emission flag data can be cited here https://doi.org/10.5067/Airborne/CAMP2Ex_TraceGas_AircraftInSitu_P3_Data_1. We reworked line 183-184 and it now reads:

*To investigate the impact of air mass type on NPF, we classified air masses sampled during CAMP2Ex into three types (details can be found at https://doi.org/10.5067/Airborne/CAMP2Ex_TraceGas_AircraftInSitu_P3_Data_1.).*

- Page 7, line 186: Here you speak about 4 regimes, but mixed urban/biomass burning is not mentioned again at any later stage in the manuscript and does not appear in Fig. 4.

Yes, the 4 regimes were defined by the reference cited above, but we simply focused on the 3 main regimes (i.e., background, biomass burning and urban emission) in the analysis. We clarified this in the revised line 193-194:

*In this study, we use the underline{first three regimes} to investigate the impact of air masses on NPF by focusing on NPF events observed in background, biomass burning and urban-influenced air masses.*

- Page 8, line 221-223: If we call these events undefined, this could be used here and would make the sentence better accessible to the reader.

We agree the sentence is a little bit long. On the other hand, we feel it is better to clearly spell out what data are used to calculate the statistics for non-NPF periods above 7.5 km here.

- Page 9, line 233: "impact" might not be the best word to use here as surface area and airmass as NPF is a result of airmass, surface area and other factors. Maybe "interplay"?

We have changed "impact" to "relationship". The sentence now reads:

*The altitude dependence of the relationships among air masses, CS, and NPF implies the competing influences from different processes (i.e., production and removal of nucleating species) that vary with altitude, which will be further discussed in Sect. 4.*

- Figure 2: As you discuss the panels in the different order in the text, it makes sense to swap the current panels b and c.

Figure 2 has been modified as suggested.

- Page 10, line 271: "was elevated compared to all the other clusters" is also true and maybe even more important.

Good point. Line 270-271 now reads:

*The NPF events classified as cluster #5 have the highest CS compared to the other clusters and were mostly observed during RF18 and RF19.*

- Page 10, line 272: These were the MT flights. Could be mentioned that those were at overall different meteorological conditions.

We edited this sentence to relate cluster #5 to the different meteorological conditions resulting from monsoon transition:

*The NPF events classified as cluster #5 have the highest CS compared to the other clusters and were mostly observed during RF18 and RF19. Both flights took place near the end of CAMP$^2$Ex during monsoon transition, when airmass origins and meteorological conditions are likely different from those of earlier flights.*

This has been corrected and now reads:

*Figure 3b shows that most of NPF occurred with high actinic flux (indicated indirectly by the UV irradiance data during this campaign), as in clusters 1, 3, and 4.*

- Page 13, line 329 – Page 14, line 344: This paragraph is difficult to follow and rather lengthy. As I suggest moving Fig. 6 to the SI, it could be shortened: The main message here is: cluster #1 has high UV, occurs at noon and has a higher CS. In contrast, cluster #2 has low UV, is at the morning and has a lower CS. Investigation of the occurrence of times with such low CS shows that such periods occur more often in the early morning than in the late afternoon and the NPF frequency for mornings is higher than for the late afternoons, which indicates that the newly formed particles are indeed formed in the early morning and are not related to NPF from the previous day.

Following the reviewer's suggestions, we have moved Fig. 6 and detailed discussion to SI. This paragraph (line 329-line 349) has been shortened and now reads:

*One possible explanation is that these new particles were formed during the previous daytime under high UV irradiance/actinic flux, survived scavenging overnight and were detected the next morning. However, the low CS conditions are much more prevalent in the early morning than in the late afternoon (Fig. S5 and related discussion). In addition, the frequency of NPF in the early morning is about 20 times higher than that in the afternoon, suggesting that new particles observed most likely formed in the morning instead of the day before. The NPF in the early morning is likely made possible by the much lower CS despite the lower UV irradiance and actinic flux. We speculate the prevalence of low surface area in the early morning is due to a combination of wet scavenging and less convection overnight.*

- Page 16, line 398-399, Fig. S4: I do not see these trends from Fig. S4. N>100 nm seems to be quite similar between the two periods. What do you mean by "as high as when NPF was absent". Please clarify. Please also change the x-axis label of the Fig. S4 (strange unit, what does 27 hours mean? Just put a normal time axis there).

The term "the other segment" should be referring to the segment where no NPF was observed. The sentence is revised to further clarify:

*For the other segment (Fig. S7, 11:15-11:25), NPF was absent and the concentrations of non-volatile particles and larger particles (> 100 nm) were three times as high as those of the NPF events (Fig. S7, 12:45-12:55).*

- Page 17, line 402: Here the focus could be more on NPF by adding "(…) and mixtures of sulfuric acid ammonia and organic vapors are shown to be efficient new particle formation agents (Lehtipalo et al., 2018, Sci. Adv.)" right after the reference to Ahern et al. (2019).

We have modified the text accordingly. It now reads:

*It remains unclear which nucleation pathway dominates particle formation observed in BB-influenced air mass, since organic vapors, ammonia (Hegg et al., 1988) and sulfuric acid can directly or indirectly originate from biomass burning plumes and contribute to formation of secondary aerosols (Ahern et al., 2019), and mixtures of sulfuric acid, ammonia and organic vapors have been shown leading to strong NPF (Lehtipalo et al., 2018).*

- Page 23, line 503: Please also refer to newer studies investigating NPF in highly polluted environments, e.g., Yao et al., 2018 (Science).

The sentence has been edited as suggested:

*The mechanism for this type of NPF is likely similar to those observed in polluted urban boundary layers (Alam et al., 2003; Zhu et al., 2014; Yao et al., 2018), where high concentrations of precursors make NPF possible despite relatively high condensation sink conditions.*

- Page 23, line 504-506: Could temperature be decisive here? Especially when organics are involved in the formation process there is an interplay between the degree of oxidation and volatility which both depend strongly on temperature. If oxidation occurs at lower altitudes and high T, highly oxygenated molecules might form which are however still not able to condense onto the smallest clusters at that temperature, but during updraft and cooling of the airmass this might become possible. This should be discussed in the manuscript. You could refer to Stolzenburg et al. 2018 (PNAS) for these competing processes.

We agree that temperature could also play an important role in the nucleation of organic species. The absence of the NPF events at low latitudes (i.e., below 3 km) is likely due to the combination of warm temperature and high condensation sink. A higher temperature leads to increased volatility of secondary organic species, slowing down the growth of the newly formed clusters. Line 504-506 has been edited and now reads:

*The absence of NPF below 2 km may result from a combination of higher CS and warmer temperature compared to those in the detrainment layers at higher altitudes. Stolzenburg et al., (2018) show that temperature impacts the growth by organics via competing processes. While a higher temperature leads to faster reaction rate and high concentration of highly oxidized molecules, it also strongly increases the volatility of organic species, therefore slowing down or even inhibiting the condensation of organic vapors onto the incipient clusters.*

[revised manuscript text omitted]

---

## Author Comment (AC2)

**Response to RC2:**

We thank the reviewers for their thoughtful and constructive comments. Please find below the response to each comment or question, including notations of improvements to the manuscript. Reviewer comments are in blue fonts and responses are in black. Changes to the manuscript are highlighted in italics.

- There is a really basic question for me, rising up in the description of instruments. From line 115 on it is explained that two particle counters are measuring number concentrations above 3 nm and 10 nm. In the following table 1, all instruments are listed and there is an addition for the second CPC, that it measures downstream of a thermodenuder and thus, measures the number concentration of non-volatile particles above 10 nm. In my view, this is a significant difference, if N10 is the total number concentration > 10 nm or the number concentration of non- volatile particles > 10 nm. This difference is crucial for all conclusions coming from this paper and thus, it is difficult to formulate a review without knowing what is investigated here. I simply assume, that the thermodenuder is not in use, i.e., not heated, otherwise the data don't make sense to me. If this is not the case and the thermodenuder was heated the interpretation has to be rewritten because it is a different parameter. I went through the individual section in more detail below.

Thanks for catching this. There were two TSI 3772 CPC deployed during CAMP$^2$Ex, one provided the concentration of total number concentration of particles with diameters greater than 10 nm ($N_{>10 \text{ nm}}$), and the other sampled downstream of a thermodenuder to provide the concentration of non-volatile particles larger than 10 nm. We thus added a line in Table 1 describing the measurements of $N_{>10 \text{ nm}}$ by TSI 3772.

| Parameter/Variable | Instruments/Methods | Sampling frequency |
|---|---|---|
| Number concentration of particles (> 10 nm) | Condensation particle counter (CPC, TSI-3772) | 1 Hz |

Section 3 Overall Statistical Analysis

- This section should be structured in a clearer way. I do not think that not all subsections fit under 'statistical analysis', e.g., the dependence of Monsoon transition. Here, another headline would help. My question is: are all subsections really needed? What are the main conclusions? The figures are not easy to understand, since every figure is slightly different.

In this section, we present an overview of the NPF events associated with different air mass types as well as the results of k-means clustering analysis, which lay a foundation for the subsequent analyses described in the next section (Section 4). The main conclusions are the increasing trend of NPF frequency with altitude and the variation of the frequency in different air masses. Following the reviewer's suggestions, we have made the following changes to this section:

1. Changed the headline to "Overview of NPF events during CAMP$^2$Ex".

2. Incorporated section 3.3 into section 3.2 and moved Fig. 1 and Fig. 3 to SI; modified Fig. S1 to show the flight tracks and locations of NPF events associated with various air mass types. Fig. S1 is now Fig. 1 in the main text.
3. Modified the figures using condensation sink instead of surface area.

Please refer to the revised manuscript for more details of the revision.

- There are some sentences, which are difficult to digest, e.g. (l. 201f.), 'Most NPF events were observed above 3 km when RH exceeded 50%, and only about 2% of total NPF was observed at 3.5-5 km.' From this, I would conclude that most NPF events occur above 5 km?! Such sentences should be homogenized.

We have edited this paragraph as suggested and it now reads:

*There was a total of 19 research flights (RFs) during CAMP$^2$Ex. Figure 1 shows an overview of the flight tracks and the locations where NPF in three major air mass types was observed. These RFs covered the ocean east and west of Luzon Island, and two of them (RF8 and RF18) sampled over Luzon Island and upwind/downwind of Metro Manila. The date and sampling areas of all RFs, together with the duration and key variables of observed NPF events, are presented in Table S1. Most NPF events were observed above 5 km when RH exceeded 50%. A few short periods with elevated $N_{>3\ nm}/N_{>10\ nm}$ (not counted as NPF events) were observed within the boundary layer about 50 kilometers downwind west of metro Manila during RF18, which are closely associated with shipping and/or urban emissions. These NPF events likely occurred immediately following the dilution of vehicle and engine emissions (e.g., Uhrner et al., 2011; Wehner et al., 2009), and they are not included in further analyses. NPF frequency, defined as the ratio of the sampling time when new particles were observed to the total flight time, decreased drastically starting from RF11 on 19 September and no events were observed from RF12 through RF17 as shown in Fig. S1. This sudden decrease in NPF frequency coincided with the early monsoon transition starting on 20 September (Hilario et al., 2021).*

- Figure 2a) it is hard to distinguish low NPF frequencies from zero. At the lowest point, it looks like the frequency in background air is significantly above zero, but it is stated above: 'Figure 2a shows that below 5.5 km, no NPF events were observed in background or BB-influenced air masses (l. 224)' This does not fit for me; such results have to be clear and comprehensible. Please check this paragraph.

Right, the NPF frequency in background air at 3.5 km is above zero. The statement has been revised in the text:

*Figure 2a shows that below 5.5 km, NPF frequency is very low (below 3%) and NPF was mostly observed in the urban-influenced air masses. No NPF events were observed in BB-influenced and only minor events took place in the background air masses at ~3.5 km.*

Fig. 2 includes all data sampled above 3 km as there were no NPF events observed below 3 km.

- Figure 3 includes the same data, but just divided into the two periods? There are too many similar looking pictures, which are not so easy to interpret.

Following the reviewer's suggestion, we have moved this figure to SI and incorporated the discussion into section 3.2 to make it concise to read.

Section 4: Characteristics of NPF in Different Air Mass Types

- In subsection 4.1 NPF events in Background air were analyzed. Here the two different clusters were shown in the following. In the subsection about biomass burning those events from a certain height region were compared with background cases. In the third subsection, the urban NPF cases were compared with non-NPF events. This is really confusing. Here, a more homogeneous way should be selected and maybe one figure containing all three airmasses, maybe in comparison with non-NPF cases can replace some of the figures.

We thank the reviewer for this comment. We tried to use a homogeneous way to compare NPF and non-NPF periods for all three air mass types but find it very challenging. This is because NPF is affected by several interplaying factors. Examining the impact of one factor on NPF often requires comparing measurements with other parameters at the same or similar level, which is often not possible given the data collected during CAMP$^2$Ex. For example, in subsection 4.2 (i.e., BB-influenced NPF), if we directly compare the measurements between NPF and non-NPF periods in BB-influenced air masses, it would be difficult to differentiate the impact of precursors (with CO as proxy) because measurements during non-NPF periods always have higher condensation sink than NPF events do. By comparing between BB-influenced NPF and background NPF, we can more clearly show the role that precursors play in enhancing NPF, given the same level of CS and UV irradiance. This is why we adopted different types of comparisons for different air mass types.

- There are definitely too many and too different figures in the manuscript and this should be changed. Also, the mix of statistical analysis, cases studies as time series and profiles is confusing and should be better structured/motivated.

We have moved a number of figures to SI, including Fig. 1, Fig. 3, Fig. 6, Fig. 9, Fig. 11 and Fig. 12. For the discussion of NPF in urban influenced air masses, time series figures (Fig. 8 and Fig. 10) are kept in the main text, whereas Figure 9 (statistical analysis) has been moved to SI to make the manuscript more concise. Please refer to the revised manuscript for the adjustments.

Summary:

- This section summarizes the most important finding of the study. However, I miss a bit more interpretation. Furthermore, it should be estimated how relevant such NPF processes are at all. Here it should be stated, which type of NPF plays a significant role and which is more a

*minor process and so on. In general, how relevant is the free tropospheric NPF for this region, for other regions? Is it possible to conclude such a statement from these results?*

We thank the reviewer for this suggestion. We used surrogates (i.e., RH, CO) for precursors from different sources and examined the impact of airmass on NPF. However, because there are no measurements of the nucleating species and their precursors (e.g., DMS, $NH_3$), we are not able to quantify the importance of different nucleation mechanisms. Understanding the role of different nucleation pathways in the FT is clearly very important, but it will require new measurements in future studies.

- *In the summary the term condensation sink is used while in figures always surface area is shown. Of course, these parameters are similar, but not identical. I would prefer if CS would be used throughout the whole paper.*

Condensation sink based on particle size distribution after hygroscopic growth has been calculated and used as one of the four parameters for the k-means classification. Detailed description of the method is added to section 2.1 (shown below), and discussions can be found throughout the manuscript.

*Condensation sink (CS) reflects how quickly condensable vapors will condense on the existing aerosol (Dal Maso et al., 2002). We calculated CS from the ambient aerosol size distribution (Kulmala et al., 2012), which was derived by combining dry particle size distribution measured by FIMS (10-600 nm) and LAS (600-1000 nm), ambient RH, and an average hygroscopicity parameter (κ). Aerosol mass spectrometer (AMS) measurements show that on average, $(NH_4)_2SO_4$ represents 90% of the $PM_1$ mass above 5 km, where the vast majority of the NPF events was identified. A κ value of 0.53 was therefore applied to calculate particle hygroscopic growth factor at ambient RH (Petters and Kreidenweis, 2007) for each size bin of the combined dry size distribution.*

- *Maybe some more comparison of the different air mass types is possible, one main difference is probably the condensation sink, but could the authors speculate also about differences in precursor gases?*

We have added a figure (Fig. S2) to SI showing the vertical profiles of CS for three major air mass types and a few sentences to section 3.2 to briefly discuss the difference:

*Figure S2 shows the vertical profiles of CS for three air mass types. Background air masses have lowest CS on average below 4 km and above 6 km (except for 7-7.5 km) among all three air mass types, whereas BB-influenced air masses have the highest CS at lower altitudes (i.e., < 4 km) and urban-influenced air masses dominates the higher altitudes (> 6 km). The condensation sinks of three air mass types are comparable in between.*

There are no measurements of precursor gases during $CAMP^2Ex$. We included discussions of possible precursors involved in different types of NPF events and relevant references (e.g., page 16, line 400-405 for BB-influenced NPF; page 23, line 502 for urban-influenced NPF).

- What about dynamics? Local mixing processes can foster nucleation and growth, is this relevant here?

Very good point. Previous studies have shown new particle formation promoted by mixing in boundary layer, lower troposphere, and tropopause regions, where vertical mixing of air parcels with different temperatures and precursor concentrations can occur. We speculate that mixing may occur in the outflow regions where the air parcels from lower altitudes are pumped aloft and mixed with the surrounding air. Such mixing could promote NPF observed during CAMP$^2$Ex. In this study, the NPF was identified using elevated $N_{>3nm}/N_{>10nm}$. As the observation of particles between 3 nm to 10 nm typically occurs several hours following the initial particle formation, we are not able to pinpoint the exact locations of initial particle formation and isolate the impact of the mixing processes on NPF.

We have added the following sentences to the 2$^{nd}$ paragraph in section 3.2 to briefly discuss the potential roles of mixing processes:

[revised manuscript text omitted]

---

## Referee Report (RR1)

The manuscript has improved, but it is still not acceptable. I cannot provide a second full review due to the lack of time, but I will raise the most important aspects from my point of view.

Measurements: there is not much said about aerosol measurements: inlet, losses, characteristics of the sampling system, conditions or pretreatment like drying and so on. Please give some more details!

Are data from the CPC after the TD used in this study? To my impression, not all data from table 1 are used in the study. They might be taken out or it should be mentioned, that they are not used here.

Section 3:

What do you mean by 'incipient particles'? A particular size range? Why do you think, they will be generally removed in clouds? Since you have the measurements, can you estimate how effective that process will be?

In the first review, I asked for more interpretation in this section, this could be improved, particularly in 3.1. and 3.2.

Please be more precise with height ranges: Sometimes it is said > 6 km, but it is meant between 3 and 6 km. Do not forget the lowest 3 km.

The summary needs to be improved and extended to conclusion. I asked in the first review for some more interpretation and did not see significant improvement here. Please add conclusions and interpretation. Good, that the dynamic/turbulent aspect was added, but dynamics does not only influence the nucleation rate (which cannot be studied with these data), but furthermore also the growth rate. This can be also a relevant factor. Is there any evidence for the dynamic impact?

The study should be thoroughly revised in terms of language. Many sentences are too long, some are worded incomprehensibly.

---

## Author Response (AR2)

We thank the editor and referees for their thoughtful and constructive comments. Please find below detailed responses to each comment or question, including notations of improvements to the manuscript. Editor and referee comments are in blue fonts. Changes and improvements to the manuscript are shown in underlined text.

**Editor comments:**

Many thanks for revising your manuscript. Referee #1 is satisfied with the changes.

However, referee #2 has some general remaining comments. Please take a look at their report and address the comments.

Please see response to comments of referee #2.

In addition, building on these comments, I would like you to further revise the following aspects:

1) Abstract

The abstract is very long. Please make it more concise. We recommend a length of 300 words while addressing the following points:

- The topic of the article and why it is important

- The status of scientific understanding

- The gap in knowledge being addressed

- The objectives, questions or hypotheses of the study

- The approach

- The main results with important quantitative information if appropriate

- The importance and implications of the results

In particular, point 3 should be added more clearly (what is the hypothesis?), whereas points 5 and 6 are addressed in a lot of detail.

We thank the editor for the suggestion. We have modified the abstract accordingly to address the points suggested above. The abstract is also more concise now (below 300 words). The revised abstract now reads:

Nucleation in the free troposphere (FT) and subsequent growth of new particles represent a globally important source of cloud condensation nuclei (CCN). Whereas new particle formation (NPF) has been shown to occur frequently in the upper troposphere over tropical oceans, there have been few studies of NPF at lower altitudes. In addition, the impact of urban emissions and biomass burning on the NPF in tropical marine FT remains poorly understood. In this study, we examine NPF in the lower and mid troposphere (3-8.5 km) over tropical ocean and coastal region using airborne measurements during the recent Cloud, Aerosol and Monsoon Processes Philippines Experiment (CAMP$^2$Ex). NPF was mostly observed above 5.5 km and coincided with elevated relative humidity (RH) and reduced condensation sink (CS), suggesting that NPF occurs in convective cloud outflow. The frequency of NPF increases with altitude, reaching ~50% above 8 km. An abrupt decrease in NPF frequency coincides with early monsoon

transition, and is attributed to increased CS resulting from reduced convective activity and more frequent transport of aged urban plumes. Surprisingly, a large fraction of NPF events in background air were observed in the early morning, and the NPF is likely made possible by very low CS despite low actinic flux. Convectively detrained biomass burning plume and fresh urban emissions enhance NPF as a result of elevated precursor concentrations and scavenging of pre-existing particles. In contrast, NPF is suppressed in aged urban plumes where the reactive precursors are mostly consumed while CS remain relatively high. This study shows strong impact of urban and biomass burning emissions on the NPF in tropical marine FT. The results also illustrate the competing influences of different variables and interactions among anthropogenic emissions, convective clouds, and meteorology, which lead to NPF under a variety of conditions in tropical marine environment.

2) Comparison to previous studies

Please note the journal scope statement that articles with a local focus must clearly explain how the results extend and compare with current knowledge. While I find several statement that you find similar features as in previous studies, I am missing a discussion on your findings extend current knowledge. Please either add brief discussions to the various subsections in section 4 or dedicate a separate section to such a discussion.

Alternatively – instead of adding such a detailed discussion – you may want to consider recategorization of your manuscript to a Measurement Report for which expects conclusions of more limited scope than in research articles.

Following the editor's suggestion, we have added discussions on findings that extend current knowledge. For example, we show that unlike regional NPF in the boundary layer that mostly occurs around noontime, in the FT over tropical oceans, strong radiation is not a necessary condition for NPF and a large fraction of the background NPF occurs under very low CS condition in the early morning when radiation and actinic flux are low. The prevalence of very low CS condition in the early morning is due to a combination of wet scavenging and less convection overnight. In addition, few direct measurements of NPF in biomass burning plumes have been reported. In this study, we found detrained biomass burning plumes strongly enhance NPF in the outflow region of convective clouds once existing particles are efficiently removed by wet scavenging. We also added a subsection discussing the impact of the urban emissions on NPF in tropical marine free troposphere:

Impact of urban emissions on NPF in tropical marine FT is poorly understood as previous measurements were mostly carried out under pristine conditions. The analyses presented above show that depending on the age and altitude, urban emissions can have different effects on NPF. Above 5.5 km (i.e., approximately the freezing level), convectively detrained fresh urban plumes have low CS due to efficient wet scavenging of existing particles, and elevated precursor concentrations contribute to and enhance NPF under the low CS condition in the outflow regions. At lower altitudes (i.e., below freezing level), NPF takes place in detrained fresh urban plumes with higher CS compared to the background. The higher CS is likely due to less efficient wet scavenging at these lower altitudes. High concentrations of precursors from the fresh urban emissions likely made these NPF possible despite relatively high CS. The species participating in these NPF events may include sulfuric acid and amines (Yao et al., 2018). Future measurements, including the precursors and nucleating species, are needed to elucidate the nucleation

mechanisms in the air masses influenced by fresh urban emissions. In aged urban plumes over 7 km, reactive precursors were mostly consumed during the long-range transport from East Asia, while CS remained relatively high. As a result, the aged urban plumes tend to inhibit NPF instead of promoting it as in the case of fresh urban emissions.

3) Conclusions

Currently, your last section is entitled 'Summary' and I think this describes its content well. However, we would expect a balanced section that also includes 'conclusions'. Please add more text on the limitations, novelties and implications of your findings to the last section that should be renamed 'Summary and Conclusions' or only the latter if you decide to shorten the current text.

We have modified the last section accordingly and renamed it as "Summary and conclusions". It now reads:

**5 Summary and conclusions**

[revised manuscript text omitted]

**Comments of referee #2:**

Measurements: there is not much said about aerosol measurements: inlet, losses, characteristics of the sampling system, conditions or pretreatment like drying and so on. Please give some more details!

We thank the referee for the suggestion. We have modified the second paragraph of Sect. 2.1 by adding details of aerosol measurements and inlet. It now reads:

The measurements examined in this study include aerosol properties, carbon monoxide (CO), methane ($CH_4$) and ozone ($O_3$) mixing ratios, meteorological parameters, and radiation (see Table 1 for details). Ambient aerosol was sampled by using a "Clarke" style forward facing shrouded solid diffuser that was operated iso-kinetically (Mcnaughton et al., 2007). Two condensation particle counters (CPCs, Model 3756 and 3772, TSI Inc.) measured the total number concentrations of particles nominally larger than ~3 and ~10 nm ($N_{>3\,nm}$ and $N_{>10\,nm}$), respectively. An additional CPC (TSI Model 3772) sampled downstream of a thermal denuder operated at 350 °C and provided non-volatile particle number concentration (nonvolatile $N_{>10\,nm}$). Aerosol size distributions were characterized by a fast integrated mobility spectrometer (FIMS, 10-600 nm; Wang et al., 2017a; Wang et al., 2017b; Wang et al., 2018) and a laser aerosol spectrometer (LAS, Model 3340, TSI Inc., 100-3000 nm). The aerosol samples measured by FIMS and LAS were dried both actively by Nafion driers and passively due to higher aircraft cabin temperature than the ambient. Size distributions provided by LAS were size-corrected assuming a particle refractive index of ammonium sulfate (Moore et al., 2021).

Are data from the CPC after the TD used in this study? To my impression, not all data from table 1 are used in the study. They might be taken out or it should be mentioned, that they are not used here.

Yes, the concentration of non-volatile particles (non-volatile $N_{>10\,nm}$) is used in Sect. 4.3.1 and Fig. 6.

Section 3: What do you mean by 'incipient particles'? A particular size range? Why do you think, they will be generally removed in clouds? Since you have the measurements, can you estimate how effective that process will be?

Incipient particles refer to the stable clusters from nucleation of gas phase species and typically have diameters between 1-2 nm. This is now clarified in the text. Given the small size of the incipient particles, they are efficiently removed by cloud droplets through coagulation. For example, in a cloud with droplet diameter of 10 μm and a number concentration of 200 $cm^{-3}$, the e-folding lifetime of 1.5 nm particles is about 90 seconds.

In the first review, I asked for more interpretation in this section, this could be improved, particularly in 3.1. and 3.2.

Please be more precise with height ranges: Sometimes it is said > 6 km, but it is meant between 3 and 6 km. Do not forget the lowest 3 km.

We added more interpretation in Section 3. For example:

"As CS is largely independent of altitude above 5.5 km (Fig. S2), the strong increase of NPF frequency is likely due to lower temperature and higher galactic cosmic rays ionization rate at higher altitudes, at least partially (Kazil et al., 2006)."

"No NPF events were observed from RF12 through RF17, despite the flight tracks overlap with the earlier flights in terms of location and altitude range. This abrupt decrease in NPF frequency coincides with the early monsoon transition starting on 20 September (Hilario et al., 2021), indicating a strong impact of synoptic condition on NPF in this region…"

"The NPF frequency during the MT is lower than that during SWM at most altitudes above 5.5 km (Fig. S3). The decrease of NPF frequency during the MT is likely due to increased CS (Fig. S3b), which may be a result of reduced convective activity as indicated by lower RH (Fig. S3c) and thus reduced wet scavenging. The more frequent long range transport of aged urban plumes may also contribute to the elevated CS during the MT (Hilario et al., 2021)."

We would also like to point out that Sections 3.1 and 3.2 describe general statistics of NPF events observed during $CAMP^2Ex$, including the vertical profiles of NPF frequency in different airmasses. The NPF in background air, and the impact of biomass burning and urban emissions on NPF in tropical marine FT are presented and interpreted in Section 4.

We have made the height ranges more precise. Below 3 km, only a few short periods with elevated $N_{>3 nm}$/$N_{>10 nm}$ were observed within the BL about 50 kilometers downwind west of metro Manila during RF18, and are closely associated with shipping and/or urban emissions. The elevated $N_{>3 nm}$/$N_{>10 nm}$ likely occurred immediately following the dilution of vehicle and engine emissions (e.g., Uhrner et al., 2011; Wehner et al., 2009), and they are not considered as NPF events and therefore excluded from further analyses. This is now clarified in the text.

The summary needs to be improved and extended to conclusion. I asked in the first review for some more interpretation and did not see significant improvement here. Please add conclusions and interpretation. Good, that the dynamic/turbulent aspect was added, but dynamics does not only influence the nucleation rate (which cannot be studied with these data), but furthermore also the growth rate. This can be also a relevant factor. Is there any evidence for the dynamic impact?

Following the referee's suggestion, we have improved the summary and extended it to conclusion. The section is now renamed as "Summary and conclusions" (please also see our response to editor's 3rd comment). We agree that local mixing processes can influence growth rate as well. Unfortunately, we don't have suitable data to examine the particle growth rate, because the flights during CAMP²Ex were not designed to track the particle evolution in the same air parcel (i.e., Lagrangian sampling). The last section now reads:

**5 Summary and conclusions**

[revised manuscript text omitted]

The study should be thoroughly revised in terms of language. Many sentences are too long, some are worded incomprehensibly.

We thank the referee for the comment. We have gone through the manuscript carefully and improved the language.